# A randomization-based causal inference framework for uncovering environmental exposure effects on human gut microbiota

**Alice J. Sommer**[1,2,3]*, **Annette Peters**[2,3,4]*, **Martina Rommel**[3,5], **Josef Cyrys**[3], **Harald Grallert**[5,6], **Dirk Haller**[7,8], **Christian L. Müller**[9,10,11]*, **Marie-Abèle C. Bind**[1,12]

**1** Department of Statistics, Harvard University, Cambridge, Massachusetts, United States of America, **2** Institute for Medical Information Processing, Biometry, and Epidemiology, Faculty of Medicine, Ludwig-Maximilians-University München, Munich, Germany, **3** Institute of Epidemiology, Helmholtz Zentrum München, Neuherberg, Germany, **4** Department of Environmental Health, Harvard T. H. Chan School of Public Health, Boston, Massachusetts, United States of America, **5** Research Unit of Molecular Epidemiology, Helmholtz Zentrum München, Neuherberg, Germany, **6** German Center for Diabetes Research (DZD), München-Neuherberg, Germany, **7** ZIEL - Institute for Food & Health, Technical University of Munich, Freising, Germany, **8** Chair of Nutrition and Immunology, Technical University of Munich, Freising, Germany, **9** Institute of Computational Biology, Helmholtz Zentrum München, Neuherberg, Germany, **10** Department of Statistics, Ludwig-Maximilians-University München, Munich, Germany, **11** Center for Computational Mathematics, Flatiron Institute, New York City, New York, United States of America, **12** Biostatistics Center, Massachusetts General Hospital and Harvard Medical School, Boston, Massachusetts, United States of America

* alice.j.sommer@gmail.com (AJS); peters@helmholtz-muenchen.de (AP); cmueller@flatironinstitute.org (CLM)

**Data Availability Statement:** The KORA cohort data discussed in the paper is available upon request via the kora.passt portal:

## Abstract

Statistical analysis of microbial genomic data within epidemiological cohort studies holds the promise to assess the influence of environmental exposures on both the host and the host-associated microbiome. However, the observational character of prospective cohort data and the intricate characteristics of microbiome data make it challenging to discover causal associations between environment and microbiome. Here, we introduce a causal inference framework based on the Rubin Causal Model that can help scientists to investigate such environment-host microbiome relationships, to capitalize on existing, possibly powerful, test statistics, and test plausible sharp null hypotheses. Using data from the German KORA cohort study, we illustrate our framework by designing two hypothetical randomized experiments with interventions of (i) air pollution reduction and (ii) smoking prevention. We study the effects of these interventions on the human gut microbiome by testing shifts in microbial diversity, changes in individual microbial abundances, and microbial network wiring between groups of matched subjects via randomization-based inference. In the smoking prevention scenario, we identify a small interconnected group of taxa worth further scrutiny, including *Christensenellaceae* and *Ruminococcaceae* genera, that have been previously associated with blood metabolite changes. These findings demonstrate that our framework may uncover potentially causal links between environmental exposure and the gut microbiome from observational data. We anticipate the present statistical framework to be a good starting point for further discoveries on the role of the gut microbiome in environmental health.

https://helmholtz-muenchen.managed-otrs.com. The code for analysis and visualization of the data are accessible on the following GitHub public repository: https://github.com/AliceSommer/Pipeline_Microbiome. A tutorial to get acquainted with the framework and open source data is accessible on the following GitHub public repository: https://github.com/AliceSommer/Causal_Microbiome_Tutorial.

**Funding:** Research reported in this publication was supported by the Office of the Director, National Institutes of Health under Award Number DP5OD021412 and the John Harvard Distinguished Science Fellows Program within the FAS Division of Science of Harvard University (MACB). The content is solely the responsibility of the authors and does not necessarily represent the official views of the National Institutes of Health. The KORA study was initiated and financed by the Helmholtz Zentrum München—German Research Center for Environmental Health, which is funded by the German Federal Ministry of Education and Research (BMBF) and by the State of Bavaria (AP). Furthermore, KORA research was supported within the Munich Center of Health Sciences (MC-Health), Ludwig-Maximilians-Universität, as part of LMUinnovativ (AP). Microbiota profiling of KORA samples was supported by enable Kompetenzcluster der Ernährungsforschung (No. 01EA1409A) and the European Union Joint Programming Initiative DINAMIC (No. 2815ERA04E, 2815ERA11E) (DH). The funders had no role in study design, data collection and analysis, decision to publish, or preparation of the manuscript.

**Competing interests:** The authors have declared that no competing interests exist.

## Author summary

Environmental influences on the human gut microbiome are still to be discovered or better understood. In this paper, we contribute to the field of microbiome research and environmental epidemiology by suggesting a stage-based causal inference framework relying on the foundations of the Rubin Causal Model. A particularity of the framework is the use of randomization-based inference, which we value to be a necessary exploratory inference method when tackling untapped research questions. To illustrate the framework, we explore the effects of two inhaled environmental exposures previously hypothesized to be linked with gastrointestinal diseases and the gut microbiome: air pollution exposure and cigarette smoking.

This is a *PLOS Computational Biology* Methods paper.

## 1 Introduction

The human microbiome plays a pivotal role in maintaining a healthy physiology via multiple interactions with the host. The gut microbiome, for instance, provides important metabolic capabilities for food digestion [1, 2] and regulates immune homeostasis [3]. Although dietary interventions [4], pathogen infections [5], and antibiotics use [6] can trigger rapid changes of gut microbial compositions and can lead to dysbiotic disruptions of host-microbiome interactions, the long-term impact of environmental exposures on the human gut microbiome remains poorly understood. In this paper, we provide a causal inference framework for assessing such epidemiological questions and analyze a prospective cohort with collected microbiome data. Recent technological advances, through culture-independent analyses, have facilitated a surge in observational studies of the human microbiome [7–9]. A common method to catalog microbial constituents is high-throughput amplicon sequencing [10], allowing the acquisition of gut microbiome survey data for large prospective cohort studies. Important examples include the Human Microbiome Project [11], the British TwinsUK study [12], the Dutch LifeLines-DEEP [13] and Rotterdam Studies [14], the Chinese Guangdong Gut Microbiome Project [15], the American Gut Project [16], and the German KORA study [17].

Thus far, these and other studies have linked alterations in gut microbial compositions to several common diseases, including rheumatoid arthritis, colorectal cancer, obesity, inflammatory bowel disease (IBD), and diabetes [18]. Although environmental exposures such as particulate matter (PM) [19] and smoking [20] are also related to these diseases, an understanding of environment-gut microbiome relationships and their implications for disease mechanisms has remained elusive. Here, we examine such environment-gut microbiome relationships within a causal inference framework [21] combined with state-of-the-art statistical methods for amplicon sequence variant (ASV) data [22]. We illustrate our analysis framework using data from the German KORA study [17] and focus on two inhaled environmental exposures previously hypothesized to be linked with gastrointestinal diseases and the gut microbiome: (i) particulate matter (PM) with diameter smaller or equal to 2.5 $\mu m$ ($PM_{2.5}$) and (ii) cigarette smoking.

Air pollution exposure has been found to be associated with gastrointestinal diseases, such as appendicitis [23], inflammatory bowel disease [24], abdominal pain [25], and metabolic

disorders [26]. Current research suggests that air pollution may impact the gut microbiome which, in turn, acts as a "mediator" of the association between air pollution and metabolic disorders such as obesity and type 2 diabetes [27–29]. These studies found associations between nitric oxide, nitrogen dioxide [27], PM [28], and ozone [30] exposures and the gut microbiome. Several potential pathways explain how particles affect human health. The gut is exposed to PM through: (i) mucociliary clearance, i.e., the self-cleaning mechanism of the bronchi, inducing inhaled PM to be cleared from the lungs to the gut, and (ii) oral route exposure, when food and water are contaminated by PM prior to being ingested or in the alimentary canal via inhalation [31, 32]. Results from murine studies of the effect of PM on the gut [33–37] suggest that exposure to PM changes the microbial composition and increases gut permeability, leading to higher systemic inflammation due to the unrestrained influx of microbial products from the gut into the systemic circulation [38].

The chemical mixture of cigarette smoke inhaled into the lungs has an effect on blood markers that, in turn, interact with the gut. Another pathway is that the toxicants of cigarette smoke swallowed into the gastrointestinal tract induce gastrointestinal microbiota dysbiosis via antimicrobial activity and regulation of the intestinal microenvironment [39]. Cigarette smoking is an inhaled exposure that has been shown to influence the susceptibility of diseases such as IBD, colorectal cancer, and systemic diseases [20, 40, 41]. Animal studies suggest that cigarette smoke may mediate its effects through alterations of intestinal microbiota [42]. In humans, shifts in the gut microbiome composition and diversity were observed after smoking cessation. These shifts were similar to previously observed shifts in obese vs. lean patients, suggesting a potential microbial link between the metabolic function of the gut and smoking cessation [43]. Comparison of the gut microbiome composition of smokers and never-smokers led to similar observations [44]. So far, the underlying mechanisms of the effect of smoking on not only gut-related, but also autoimmune diseases have not been established. It has been hypothesizes that the gut microbiome may be the missing link between smoking and autoimmune diseases [20].

Central to the present study is the investigation of the causal question: *Does reducing inhaled environmental exposures alter the human gut microbiome?* As summarized in Fig 1, we answer this question using the following four-stage analysis framework: (i) conceptualize hypothetical environmental interventions that could have resulted in the observed data at hand, (ii) design our non-randomized data, so that the unconfoundedness assumption can be assumed, (iii) choose powerful, state-of-the-art test statistics from the literature to compare human gut microbiome at different levels of taxonomic granularity between subjects assigned to the interventions vs. not, and (iv) interpret the implications of the results for recommending further studies or the studied hypothetical intervention. The reason for using this four-stage approach is for the transparency of its assumptions when interpreting results. The Methods section elaborates on each of these steps. An essential ingredient in stage (iii) of our framework is the use of a randomization-based hypothesis testing with powerful test statistics comparing subjects under an intervention vs. not [45, 46]. We do not attempt to provide an estimate of (and uncertainty around) an estimand to avoid relying on assumptions such as the additivity of the treatment effects, asymptotic arguments, or an imputation model, which may be the case when drawing Neymanian (i.e., distribution-based) or Bayesian inferences. This Fisherian approach is a non-asymptotic first step to start shedding light on merely-touched research questions dependent on complex data structures, such as human gut microbiome data.

The present causal inference framework relies on ideas developed in the 70s [47–50] and the Rubin Causal Model [51, 52] to analyze observational data by reconstructing the ideal conditions of randomized experiments, the "gold standard" to draw objective causal inferences on the effects of an intervention [53]. A formidable statistical challenge is, however, to define and

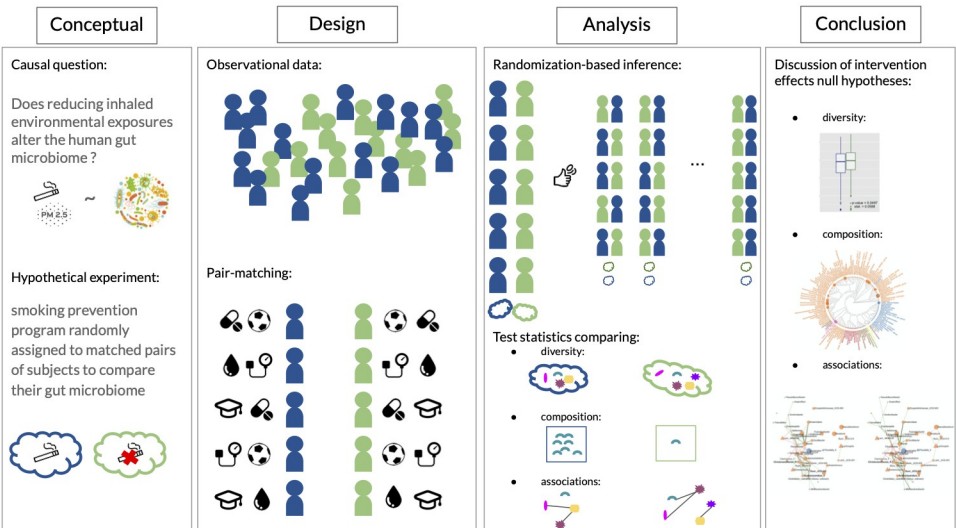

**Fig 1. The four stages of the causal inference framework [21] adapted to the exploration of environment-gut microbiome relationships.** Stage 1: Formulation of a plausible hypothetical intervention (e.g., decreasing inhaled environmental exposures) to examine its impacts on the gut microbiome. Stage 2: Construct a hypothetical paired-randomized experiment in which the environmental intervention been implemented randomly. Stage 3: Choose powerful test statistics comparing the gut microbiome had the subjects been hypothetically randomized to the environmental intervention vs. not and test the sharp null hypotheses of no effect of the intervention at different aggregation levels of the data. Stage 4: Interpretation of the statistical analyses and recommendations for future studies or implementation of the intervention.

test these intervention effects for high-dimensional taxonomically-structured microbiome relative abundance data. Here, we adapted and advanced several state-of-the-art approaches from the statistical literature tailored to amplicon data, ranging from tests for $\alpha$-diversity in networked communities [54, 55], Microbiome Regression-based Kernel Association Tests (MiRKAT) for $\beta$-diversity to randomization-based differential compositional mean tests [56]. We also applied and analyzed individual taxon differential abundance tests with taxonomic rank-dependent reference selection [57] and sparse compositionally robust taxon-taxon network estimation schemes [58] with novel differential edge tests [59], thus covering a comprehensive list of archetypical microbiome data analysis tasks.

Our framework complements recent causal inference approaches for microbiome data such as mediation methods [60, 61], graphical models [62], and Mendelian randomization [63, 64] to analyze observational gut microbiome data. In these studies, the target for interventions is the microbiome and the understanding of its effects on diseases, i.e., the microbiome is treated as the exposure and diseases as outcomes. Here, we are interested in examining the effects of environmental exposures (interventions) on the gut microbiome ("the" outcome), when only non-randomized data are available. To the best of our knowledge, no other observational study interested in environmental effects on the gut microbiome addressed their research question using causal inference methods.

In the following, we detail the characteristics of the KORA FF4 study population and highlight potential effects of the hypothetical interventions, air pollution reduction and smoking prevention, on the gut microbiome. In particular, we characterize potential effects in terms of changes in overall microbial diversity, taxon-level abundances, and microbial associations. In the smoking prevention analysis, we identified taxa, including *Ruminococcaceae (UCG-005,*

*UCG-003, UCG-002)* and *Christensenellaceae R-7-group*, that are part of a stable sub-community in the microbial association networks and have been found to contribute to circulating blood metabolites in the LifeLines-Deep cohort [65].

## 2 Methods

### 2.1 The German KORA FF4 cohort study

The data come from the German KORA FF4 cohort study, which involves participants aged 25 to 74 years old living in the city of Augsburg [17]. The participants were subject to health questionnaires and follow-up examinations. During the study, stool samples were collected and the gut microbiota data for 2,033 participants were obtained with 16S rRNA gene sequencing. For each participant we have their long-term exposure to air pollution (particulate matter). The long-term exposure variables come from the ULTRA III study, in which air pollutants were monitored several times a year at 20 locations within the Augsburg region. From this data, annual averages of air pollutants were calculated using land-use regression models. The models explain the spatial variation of the pollutants with predictor variables derived from geographic information systems (GIS). To obtain the long-term air pollution values for each participant, land-use regression models were applied to their residential address. Moreover, to elucidate relationships between health outcomes and diet, dietary intake data were collected for 1,469 participants of the KORA FF4 cohort. Dietary intake was derived using a method combining information from a food frequency questionnaire (FFQ) and repeated 24-h food lists [66]. In brief, the usual food intake (in gram/day) was calculated as the product of the probability of consumption of a food on a given day and the average amount of a food consumed on a consumption day.

**2.1.1 Gut microbiome data sequencing and preprocessing. DNA Extraction, 16S rRNA Gene Amplification, and Amplicon Sequencing**. Fecal DNA extraction was isolated by following the protocol of [67]. The samples were profiled by high-throughput amplicon sequencing with dual-index barcoding using the Illumina MiSeq platform. Based on a study providing guidelines for selecting primer pairs [68], the V3-V4 region of the gene encoding 16S ribosomal RNA was amplified using the primers 341-forward (CCTACGGGNGGCWGCAG; bacterial domain specific) and 785-reverse (GACTACHVGGGTATCTAATCC; bacterial domain specific). Amplification was undertaken using the Phusian High-Fidelity DNA Polymerase Hotstart as per manufacturer's instructions. The PCR libraries were then barcoded using a dual-index system. Following a round of purification with AMPure XP beads (Beckman Coulter), libraries were quantified and pooled to 2nM. The libraries were sequenced on an Illumina MiSeq (2 x 250 bp), using facilities provided by the Ziel NGS-Core Facility of the Technical University Muenchen (TUM).

**Bioinformatics**. The demultiplexed, per-sample, primer-free amplicon reads were processed by the DADA2 workflow [22, 69] to infer sequence variants, remove chimeras, and assign taxonomies with the Silva v128 database [70] using the naive Bayesian classifier method [71] until the genus-level assignment and the exact matching method [72] for species-level assignment. We opted for the high-resolution DADA2 method to infer sequence variants without any fixed threshold, thereby resolving variants that differ by as little as one nucleotide. Amplicon sequence variants (ASVs) do not impose the arbitrary dissimilarity thresholds that define OTUs. They provide consistent labels because they represent a biological reality that exists outside the data being analyzed: the DNA sequence of the assayed organism, thus they remain consistent into the indefinite future [22]. The result of the DADA2 pipeline is two datasets: (i) a ASV count dataset, where each row specifies how often an ASV was sequenced and (ii) a taxonomic assignment dataset, where each row specifies the taxonomic names of an

ASV. It is common to create a phylogenetic tree of the ASVs to later on calculate microbial diversity measures such as the DivNet [55] and UniFrac [73] (see the Statistical analysis stage of Methods Section 2). The multiple genome alignment for the phylogenetic tree was built with the `DECIPHER` R package enabling a profile-to-profile method aligns a sequence set by merging profiles along a guide tree until all the input sequences are aligned [74]. The multiple genome alignment was used to construct the *de novo* phylogenetic tree using `phangorn` R package. We first construct a neighbor-joining tree [75], and then fit a maximum likelihood tree using the neighbor-joining tree as a starting point. After 16S rRNA sequencing the 2,033 stool samples from the KORA cohort and processing the sequences with the DADA2 pipeline, we observe 15,801 ASVs (see Fig A and Table A in S1 Text).

## 2.2 Causal inference framework

The four stages of the causal framework [21] that we use to construct hypothetical randomized experiments to study the environment-microbiome relationship are the following:

1. *Conceptual*: Formulation of a plausible hypothetical intervention (e.g., decreasing air pollution levels) to examine its impacts on the gut microbiome.

2. *Design*: Reconstruct the hypothetical randomized experiment had the environmental intervention been implemented randomly.

3. *Analysis*: Choose valid and powerful test statistics comparing the gut microbiome had the subjects been hypothetically randomized to the environmental intervention vs. not and test the sharp null hypotheses of no effect of the intervention at different aggregation levels of the data.

4. *Summary*: Interpretation of the statistical analyses and recommendations for future studies and interventions.

## 2.3 Conceptual stage: Formulation of the hypothetical randomized experiment in terms of potential outcomes

To understand whether environmental interventions have an effect on the human gut microbiome, the objective is to reconstruct a hypothetical experiment that mimics a controlled randomized experiment [53], in which an environmental intervention could be believed to have been randomized. Let $W_i$ be the indicator of the assignment for subject $i$ ($i = 1, \ldots, N$) to an environmental intervention vs. none, where:

$$W_i = \begin{cases} 1 & \text{if } i \text{ is under the intervention,} \\ 0 & \text{if } i \text{ is not.} \end{cases} \tag{1}$$

The composition of a human gut microbiome can be expressed as a $B$-dimensional vector of the microbial abundance. We define $Y_i^b$ as the real abundance (count) of the $b^{th}$ bacterial taxon, $b = 1, \ldots, B$ for subject $i$. We define the potential outcomes of subject $i$ as $Y_i^b(1)$, the $b^{th}$ taxon abundance (count) had subject $i$ been randomized to the environmental intervention ($W_i = 1$), and $Y_i^b(0)$, had subject $i$ not been randomized to the intervention ($W_i = 0$). Table 1 shows the potential outcomes for the $N$ subjects.

Only one of the two potential outcomes can actually be observed for each subject: this is why the Rubin Causal Model characterizes causal inference as a *missing data problem* [52], where the observed outcome of subject-$i$ and taxa-$b$ can be expressed as a function of both

**Table 1. Potential outcomes for the subjects of the hypothetical experiment.**

| Taxa | 1 | | 2 | | ... | | B | |
|---|---|---|---|---|---|---|---|---|
| Subjects | $W_i = 0$ | $W_i = 1$ | $W_i = 0$ | $W_i = 1$ | | | $W_i = 0$ | $W_i = 1$ |
| 1 | $Y_1^1(0)$ | $Y_1^1(1)$ | $Y_1^2(0)$ | $Y_1^2(1)$ | ... | ... | $Y_1^B(0)$ | $Y_1^B(1)$ |
| 2 | $Y_2^1(0)$ | $Y_2^1(1)$ | $Y_2^2(0)$ | $Y_2^2(1)$ | ... | ... | $Y_2^B(0)$ | $Y_2^B(1)$ |
| ... | ... | ... | ... | ... | ... | ... | ... | ... |
| N | $Y_N^1(0)$ | $Y_N^1(1)$ | $Y_N^2(0)$ | $Y_N^2(1)$ | ... | ... | $Y_N^B(0)$ | $Y_N^B(1)$ |

potential outcomes:

$$Y_i^{b,obs} = W_i Y_i^b(1) + (1 - W_i) Y_i^b(0) \tag{2}$$

**2.3.1 Observed outcomes measurement.** The human gut microbiome can be composed of trillions of bacteria. However, due to technology limitations, the exact abundance and number of all strains present in a human subject cannot be measured. To tackle this limitation, we opted for the processing of Amplicon Sequence Variants (ASVs) from our sequencing data to approximate the true gut microbiome composition of our study population [22, 69]. ASVs refer to individual DNA sequences recovered from a high-throughput marker gene analysis, the 16S rRNA gene in our case. Therefore, in this study the observed outcome under investigation is a $N \times A$ matrix, for $a = 1, \ldots, A$ ASVs, an approximation of the $N \times B$ matrix described above. This limitation adds another layer of missing data, i.e., we are missing the true gut microbial composition of each subject. We define the ASV counts we measured for each subject-$i$ as $C_i^{a,obs}$, which corresponds to $Y_i^{b \in A, obs}$ plus some measurement error.

## 2.4 Design stage: Reconstruction of the conceptualized hypothetical experiment

To assess causality, randomized experiments have long been regarded as the "gold standard". We are interested in the effect of environmental interventions that are often unpractical or ethical to assign randomly to humans within an experiment [21]. Therefore, we resort to a design stage [76] with a matched-sampling strategy to construct two hypothetical randomized experiments to assess the effects of an intervention on the changes in gut microbiome composition. The aim of our pair-matching strategy is to achieve balance in background covariates distributions as it is expected, on average, in randomized experiments. This approach attempts to create exchangeable groups as if the exposure was randomly assigned to each participant given measured covariates, to guarantee exposure assignment is not confounded by the measured background covariates. The exposure assignment mechanism determines which units receive which exposure; in other words, which potential outcomes are observed and which are missing [52]. The unconfoundedness of the assignment mechanism given covariates is a key assumption of the Rubin Causal Model.

Our pair-matching strategy aims to remove individual-specific confounding (e.g., years of age, sex, unit of BMI). Briefly, subject $i$ under $W_i^{obs} = 1$ with pre-exposure covariates $\mathbf{X}_i$ is matched to subject $i^\star$, under $W_{i^\star}^{obs} = 0$ only if $\mathbf{X}_{i^\star}$ is "similar" to $\mathbf{X}_i$. For each unit, the vector of covariates is given by $\mathbf{X}_i = (X_i^{(1)}, \ldots, X_i^{(k)})$. In order to ensure covariate balance, we only allow a treated unit to be matched with a control unit if the component-wise distances between their covariate vectors are less than some pre-specified thresholds $\delta_1, \ldots, \delta_k$. For any pair of

covariate vectors $X_i$ and $X_{i^\star}$, we define the difference between them as

$$\Delta(X_i, X_{i^\star}) = \begin{cases} 0 & \text{if } |X_i^{(k)} - X_{i^\star}^{(k)}| < \delta_k \text{ for } k = 1, \ldots K, \\ +\infty & \text{otherwise} \end{cases} \tag{3}$$

This constrained pair matching can be achieved using a maximum bipartite matching [77] on a graph such that: (i) there is one node per unit, partitioned into intervention nodes and control nodes, (ii) the edges are pairs of treated and control nodes with covariates $X_i$ and $X_{i^\star}$, and (iii) an edge exists if and only if $\Delta(X_i, X_{i^\star}) < +\infty$. By construction, using a maximum bipartite matching algorithm on this graph as implemented in the `igraph` R package produces the largest set of matched pairs that satisfy the unit-specific proximity constraints set by our thresholds. Let $N_E = \sum_{i=1}^{N} W_i$ be the number of subjects under the environmental intervention and $N_C = \sum_{i=1}^{N} 1 - W_i$ the number of control subjects, after matching.

After excluding the participants of the cohort that take antibiotics and had a cancer of the digestive organ, the pre-matched data set consists of 1,967 participants. At this stage, the objective is to create balanced data subsets for which the plausibility of the "unconfoundedness" assumption is based on a diagnostic of our choice. We choose the thresholds, $\delta_1, \ldots, \delta_7$, according to the pre-matching diagnostic plots of the covariate distributions (see Figs B-G in S1 Text). We privilege a large dataset with balance, while assuring that the created pairs, or in other words "twins", are scientifically plausible, e.g., no male and female could be matched. We assume a covariate to be balanced when its distribution is approximately the same under the exposure vs. not. The thresholds are: the absolute differences between the amount of alcohol consumption is less than $\delta_1 = 25$ g/day, between the body-mass-index is less than $\delta_2 = 4$ kg/m$^2$, between age is less than $\delta_3 = 5$ years, the diabetes status (diabetic, non-diabetic) is identical, i.e., $\delta_4 = 0$, and so are sex (male, female), i.e., $\delta_5 = 0$, and physical activity (active, inactive), i.e., $\delta_6 = 0$. Additionally, in the air pollution reduction experiment: the smoking status (smoker, ex-smoker, never-smoker) is identical, i.e., $\delta_7 = 0$, and in the smoking prevention experiment: the absolute difference between years of education is less than $\delta_7 = 3$ years.

After matching, we obtain two subsets of the data that can be analyzed as coming from two pair-randomized experiments: (i) an air pollution (ap) reduction hypothetical experiment ($N_{ap} = 198$), and (ii) a smoking prevention hypothetical experiment ($N_s = 542$); both data sets exhibit no evidence against covariate imbalance (see Table 2 and Figs B-G in S1 Text).

It is well known that diet has an influence on the gut microbiome and future studies on the gut should include dietary intake data in their analysis [78, 79]. In our study, we only have access to dietary intake data for a portion of our samples, therefore we examine balance diagnostics in usual nutrient intake after matching in order to maintain a large data set before matching. Figs H-I in S1 Text show that after matching, our intervention and control units (in both hypothetical experiments) do not exhibit imbalance with respect to the following food items: potatoes/roots, vegetables, legumes, fruits/nuts, dairy products, cereal products, meat, fish, egg products, fat, and sugar. In the same way, we checked for covariate balance after matching for medication intake, also a well-known confounder in human gut microbiome studies. Figs D and G in S1 Text show that after matching, our intervention and control units (in both hypothetical experiments) do not exhibit imbalance with respect to medication intake.

## 2.5 Statistical analysis stage: Randomization-based inference

To compare the gut microbiome of subjects under the environmental intervention to control subjects, we choose to not rely on asymptotic arguments, but instead take a Fisherian

**Table 2. Before and after matching number of units.** The thresholds for the air pollution experiment are based on $90^{th}$ and $10^{th}$ percentiles of the $PM_{2.5}$ distribution.

| | Air pollution | | Smoking | |
|---|---|---|---|---|
| | $N_C$ | $N_E$ | $N_C$ | $N_E$ |
| *Matching* | $PM_{2.5} \geq 13.0\ \mu g/m^3$ | $PM_{2.5} \leq 10.3\ \mu g/m^3$ | Smoker | Never smoker |
| Before | 206 | 193 | 302 | 908 |
| After | 99 | 99 | 271 | 271 |

perspective (i.e., randomization-based inference) [45, 80]. We test sharp null hypotheses ($H_0$) of no effect of the intervention for any unit by choosing test statistics that account for the complex microbiome data structure, including the additional "layer" of missing data. The ASV count data has a challenging structure because: (i) it is high-dimensional, (ii) some ASVs have low prevalence, (iii) the ASVs are strongly correlated, and (iv) it is compositional. ASV-count data is said to be "compositional" because between units comparison of ASV counts might not be informative due to the limited sequencing depth of the machine and the total number of sequenced reads varies from unit to unit (i.e., they have no common denominator) [81].

In randomization-based inference the goal is to construct the null randomization distribution of a test statistic assuming $H_0$, $T$, by computing the values of the test statistic for all possible intervention assignments. Because the number of assignments is very large, we calculate an approximating p-value using $N_{iter}$ iterations, i.e., the proportion of computed test statistics that are as large or larger than the observed test statistic: $\frac{1}{N_{iter}} \sum_{l=1}^{N_{iter}} \mathbb{1}_{T_l \geq T^{obs}}$, where $\mathbb{1}_{T_l \geq T^{obs}} = 1$ when $T_l \geq T^{obs}$, and 0 otherwise (for two-sided tests we obtain the p-values by taking absolute value of $T_l$ and $T^{obs}$, i.e., $|T_l|$ and $|T^{obs}|$). A small p-value shows that the observed test statistic is a rare event when the null hypothesis is true, which indicates the results are worth further scrutiny [82]. In the following subsections, we describe the null hypotheses we test and the test statistics we use to draw randomization-based inferences with $N_{iter}$ = 10,000 possible intervention assignments following a matched-pair design (see summary Table 3). This means that the permutations of the intervention assignment vectors needed to calculate the Fisher p-values follow the design of our hypothetical experiments. When units have varying probabilities of being treated, the analysis of experiments, even when hypothetical, should reflect their design [53, 76].

**2.5.1 Diversity analyses.** *Within Subjects Diversity.

One of the challenges of analyzing ASV-count data is working around the low prevalence of some ASVs that are due to the limited sequencing depth of the machine and the fact that some ASVs are not shared in the entire population (see Fig A in S1 Text). Therefore, before directly testing within-subject diversity differences with so called "plug-in" estimates, it has been recently suggested to start with estimating the diversity with statistical models [54]. We will

**Table 3. Data transformation and choice of test statistics.**

| analysis level | data transformation | test statistic |
|---|---|---|
| richness | breakaway [83] | betta regression coefficient [54] |
| $\alpha$-diversity | DivNet [55] | betta regression coefficient [54] |
| $\beta$-diversity | pairwise distance matrices | MiRKAT score statistic [84] |
| high-dimensional means | centered log ratios | mean abundance difference [56] |
| abundance | normalization by ratio [57] | LogFold mean difference |
| correlation | association matrices [58] | differential associations [59] |

follow this idea by estimating richness with the breakaway method [83] and estimating the Shannon index for $\alpha$-diversity with the DivNet method [55].

**Richness**. The sharp null hypothesis of no effect of the intervention on the richness can be written as: $\mathbf{H_{0,R}} : \sum_{b=1}^{B} \mathbb{1}_{Y_i^b(0)>0} = \sum_{b=1}^{B} \mathbb{1}_{Y_i^b(1)>0}$. To estimate the richness of subject $i$ (i.e., the number of bacterial taxa present in subject $i$), we will estimate the total richness in subject $i$, observed and unobserved, by $B_i$ with the `breakaway` model [83]. Let $f_{i,1}, f_{i,2}, \ldots$ denote the number of bacterial taxa observed once, twice, and so on, in a subject $i$, and let $f_{i,0}$ denote the number of unobserved bacteria, so that $B_i = f_{i,0} + f_{i,1} + f_{i,2} + \ldots$. The idea behind the breakaway method is that for each subject $i$, it predicts the number of unobserved bacteria, $f_{i,0}$, with a non-linear regression model to, in turn, provide an estimate of $B_i$.

**$\alpha$-diversity**. The sharp null hypothesis of no effect of the intervention on $\alpha$-diversity can be written as: $\mathbf{H_{0,\alpha}} : \sum_{b=1}^{B} Y_i^b(0) = \sum_{b=1}^{B} Y_i^b(1)$. To have estimates for indices of the $\alpha$-diversity of subject $i$ (i.e., its total microbial abundance) and their variance, we use the `DivNet` method, because it accounts for the co-occurrence patterns (i.e., ecological networks) of bacterial taxa in the microbial community [55]. Let $Z_i^b = Y_i^b / \sum_{b=1}^{B} Y_i^b \in [0, 1]$ denote the unknown relative abundance of taxa $b$ in subject $i$, noting that $\sum_{b=1}^{B} Z_i^b = 1$. As a reminder, $C_i^{a,obs}$ denotes the number of times taxa $a$ was observed in the stool sample of subject $i$ in our data. One of the most common $\alpha$-diversity indices is the Shannon entropy [85], which is defined as: $\alpha_{i,Shannon} = -\sum_{b=1}^{B} Z_i^b \; log(Z_i^b)$. This index captures information about both the species richness (i.e., number of species) and relative abundances of the species: as the number of species in the population increases, so does the Shannon index, and as the relative abundances diverge from a uniform distribution and become more unequal, the Shannon index decreases. In the ecological literature, researchers mostly use the following maximum likelihood estimate of $\alpha_{i,Shannon}$ (often referred to as a "plug-in" estimate): $-\sum_{a=1}^{A} \frac{C_i^a}{\sum_{a=1}^{A} C_i^a} \; log\left(\frac{C_i^a}{\sum_{a=1}^{A} C_i^a}\right)$. It has been proven that this estimate is negatively biased [86]. Therefore, various corrections have been proposed and are detailed in [55]. However, most of the suggested estimates are only functions of the ASV count vectors $C_i^a$ and do not utilize the full ASV count data matrix $C$ and the co-occurrence pattern, i.e., ecological network, of the ASVs. Willis and Martin [55] showed that these networks can have substantial effects on estimates of diversity and proposed an approach, called DivNet, to estimating diversity in the presence of an ecological network. DivNet estimates are based on log-ratio transformations by fixing a "baseline" taxon for comparison, which are modeled by a multivariate normal distribution to incorporate the co-occurrence structure between the taxa as the covariance matrix. The main advantage of DivNet method is the use of information shared across all samples to obtain more precise and accurate estimates.

**Choice of test statistic**. The test statistic we use to test $\mathbf{H_{0,R}}$ and $\mathbf{H_{0,\alpha}}$ are the coefficient of the intervention indicator estimated by the regression suggested by Willis et al. [54]. Using the coefficient of a model as the test statistic of a Fisher test was introduced in the 70s [87]. At this stage, to achieve larger bias reductions, frequentist regression models can be used to remove residual confounding that was not accounted for, during the design stage [47, 48].

Willis et al. [54] suggest to test changes in richness ($B_i$) and $\alpha$-diversity ($\hat{\alpha}_i$) with a hierarchical regression model, assuming that richness is a function of: the intervention indicator $W_i$, random variation that is not attributed to the covariates, and the standard error previously estimated with breakaway or DivNet (because not every bacterial taxon in each subject was observed so we cannot not know the true richness or $\alpha$-diversity for any $i$). The regression models are built with the `betta` function available in the `breakaway` R package [54, 83].

*Between Subjects Diversity*.

**$\beta$-diversity**. Distance-based analysis is a popular approach for evaluating the association between an exposure and microbiome diversity. The pairwise distances, $d_{ii^*}$, for high-dimensional data we consider are the: UniFrac (unweighted) distance [73], Jaccard index, Aitchison distance [88] (i.e, Euclidean distance on centered log-ratio transformed data), and Gower distance [89] (on centered log-ratio transformed data). We choose the unweighted paired UniFrac, because it is a distance metric (i.e., a non-negative real-valued function) as opposed to the generalized UniFrac. In the same way, the Jaccard distance was chosen as opposed to the commonly used Bray-Curtis. The sharp null hypothesis of no effect of the intervention on $\beta$-diversity can be written as: $\mathbf{H_{0,\beta}}$: $\mathbf{d}_{ii^*}(0) = \mathbf{d}_{ii^*}(1)$.

**Choice of test statistic**. Despite the popularity of distance-based approaches, the field of microbiome studies suffers from technical challenges, especially in selecting the best distance. Therefore, we use the suggested microbiome regression-based kernel association test (MiRKAT) [84] that uses a kernel regression and a standard variance-component score test statistic [90]. To consider different distance measures, the optimal MiRKAT: tests $\mathbf{H_{0,\beta}}$ for each individual kernel, obtains the p-value for each of the tests, and then adjust for multiple comparison with a p-value with an omnibus test. Instead, we use a fully randomization-based multiple comparison adjustment method detailed subsequently.

**Multiple comparison adjustments**. We follow the fully randomization-based procedure for multiple comparisons adjustments suggested by Lee et al. [91], which is directly motivated by the intervention assignment actually used in the experiment. This procedure has been suggested to have sufficient power to detect causal effects [91]. In our hypothetical experiments, we have matched paired intervention assignments. Both the unadjusted and adjusted p-values in the procedure are randomization-based, so do not require any assumptions about the underlying distribution of the data. The *adjusted* p-values are calculated following Steps 1–4:

1. Calculate for each hypothesis *h*, an unadjusted p-value for the observed test statistic by taking the proportion of computed test statistics that are as large or larger than the observed test statistic. This procedure is detailed in the introduction of the Statistical analysis stage section. Also, for each hypothesis *h*, $h = 1, .., H$, and intervention assignment iteration *iter*, $iter = 1, \ldots, N_{iter}$, record the vector of calculated test statistics $T_\beta^{h,iter} = (T_\beta^{1,1}, \ldots, T_\beta^{H,N_{iter}})$.

2. For each *h* and each iteration *iter*, calculate an unadjusted randomization-based p-value, with $T_\beta^{h,iter}$ as the observed test statistic. For each *iter*, record the minimum p-value of the *H* p-values.

3. The repetitions of Step 2 capture the joint randomization distribution of the test statistics and thus, of the unadjusted p-values.

4. To calculate the adjusted p-values for the observed test statistics, for each *h*, take the proportion of "minimum p-values" (recorded in Step 2) that are less than or equal to its unadjusted p-value calculated in Step 1.

Step 2–3. essentially represent a translation of the multiple test statistics into p-values sharing a common 0–1 scale.

**2.5.2 Composition analyses.** *Compositional equivalence.*

The compositionality problem means that: a change in abundance (i.e., sequenced counts) of a taxon in a sample induces a change in sequenced counts across all taxa. This problem, among others, leads to many false positive discoveries when comparing taxon abundances between groups. Moreover, because the components of a composition must sum to unity, directly applying standard multivariate statistical methods intended for unconstrained data to compositional data may result in inappropriate and misleading inferences [88]. Therefore, we

impose a centered log-ratio transformation of the compositions before testing the null hypothesis of no difference in average microbial abundance as suggested by [56].

For the measured microbiome data $C$, the centered log-ratio matrices $L = (L_i, \ldots, L_N)$ are defined by $L_i^a = log\left(\frac{C_i^a}{g(\mathbf{C}_i)}\right)$, where $g(\mathbf{C}_i) = \left(\prod_{a=1}^A C_i^a\right)^{1/A}$ denotes the geometric mean of the vector $\mathbf{C}_i = (C_i^1, \ldots, C_i^A)$. The sharp null hypothesis of no microbiome composition difference between the subjects under the intervention vs. not can be written as $\mathbf{H_{0,M}}$: for each subject $i$, $L_i(0) = L_i(1)$.

**Choice of test statistic**. The scale invariant test statistic suggested by [56] for testing $\mathbf{H_{0,M}}$ is based on the differences $\bar{L}_E^{a,obs} - \bar{L}_C^{a,obs}$, where $\bar{L}_E^{a,obs} = 1/N_E \sum_{i:W_i=1} L_i^a$ is the sample mean of the centered log ratios for subjects under the intervention. Because microbiome data are often sparse (i.e., only a small number of taxa may have different mean abundance), the following test statistic is considered: $T_M = \frac{N_E N_C}{N_E + N_C} \max_{1 \le a \le A} \frac{(L_E^{a,obs} - L_C^{a,obs})^2}{\hat{\gamma}_{aa}}$, where $\hat{\gamma}_{aa}$ are the pooled-sample centered log-ratio variances.

*Differential abundance*

The compositional nature of the microbiome data requires to choose appropriate reference sets with respect to which testing of changes in individual taxon relative abundances becomes feasible [81]. A recent approach that follows this methodology is the DACOMP (**d**ifferential **a**bundance testing with **comp**ositionality adjustment) method, proposed by [57]. DACOMP is a data-adaptive approach that: 1) identifies a subset of non-differentially abundant (reference) ASVs ($R$) in a testing dataset, and 2) tests the null of no differential abundance (DA) of the other ASVs ($a$) "normalized-by-ratio" in a training dataset. First, a taxon enters the set $R = (r_1, \ldots, r_F)$ if it has low variance ($< 2$) and high prevalence ($> 90\%$) (see Figs L-M in S1 Text). For the analyses at the ASV level, we chose the variance to be $< 3$ and the prevalence to be $> 40\%$ as thresholds in order the have at least one reference per subject. Second, using the suggested "normalization-by-ratio" approach, the null hypothesis to be tested for ASV $a$ is that ASV $a$ is not differentially abundant: $\mathbf{H_{0,DA}^{(a \notin R)}} : \frac{C_i^a(0)}{C_i^a(0) + \sum_{f=1}^R C_i^{r_f}(0)} = \frac{C_i^a(1)}{C_i^a(1) + \sum_{f=1}^R C_i^{r_f}(1)}$,

**Choice of test statistic**. To test this sharp null hypothesis, we use the LogFold change available in the `dacomp` package with the `Compute.resample.test` function. This function is useful to perform randomization-based inference for differential abundance testing, because it enables to directly incorporate a matrix of hypothetically randomized intervention assignments, which is an appealing feature when researchers work with particular designs. Because we are testing $\mathbf{H_{0,DA}^{(a \notin R)}}$ $||A|| - ||R||$ times at all taxonomic ranks, we adjust for multiple tests with the method described in the $\beta$-diversity analysis section [91].

*Partial correlation structure*

For our matched intervention and control subjects, we predicted microbial association networks using the Sparse InversE Covariance estimation for Ecological ASsociation Inference (SPIEC-EASI) framework [58] that uses 1) centered log-ratio transformations of the observed ASV counts, $C_i^{a,obs}$, to perform 2) Sparse Inverse Covariance selection (with the graphical lasso method [92]), and finally 3) pick a model based on edge stability (with the StARS method [93]) to obtain a sparse inverse covariance matrix. The non-zero entries of this matrix are proportional to the negative partial correlations among the taxa and form the edge set in an undirected weighted graph $G = (V, E)$. Here, the vertex (or node) set $V = v_1, \ldots, v_p$ represents the $p$ genera and the edge set $E \subset V \times V$ the possible associations among them. The null hypotheses of no effect of the environmental intervention on the observed genera network associations can be expressed as: $\mathbf{H_{0,N}}$: $E(0) = E(1)$.

**Choice of test statistic**. We compare the intervention and control networks with test statistics for the difference in genera associations individually. To generate sampling distributions of the test statistics under $H_{0,N}$, the intervention and control labels are reassigned 10,000 times to the samples while the matched pair structure is maintained, i.e., the assignment to intervention or control is permuted within each pair. The SPIEC-EASI framework is then re-applied to each permuted data set. This procedure is implemented with the Network Construction and Comparison for Microbiome Data, `NetCoMi`, R package [59]. To adjust for multiple differential association tests, we use the method described in the $\beta$-diversity and differential abundance analyses section [91].

## 2.6 Summary stage: Interpretation of the results

If the null hypothesis of no difference in the gut microbiome between the matched groups of treated and control units is rejected, that difference warrants further scrutiny to assess whether it can be attributed to the different treatments, assuming the assignment "unconfoundness" assumption holds. We can then report that the gut microbiome composition was or was not altered by the introduction of the environmental intervention. It is important to note that interpretation should be restricted to units that remain in the finite sample after matching (see their detailed characteristics in Figs B-I in S1 Text). The data do not provide direct information for "unmatched" units. Caution regarding extrapolation to units with covariate values beyond values observed in the balanced subset of the data is necessary.

## 3 Results

To illustrate our causal inference framework, we first conceptualize two hypothetical environmental interventions that potentially influence the gut microbiome: (i) an air pollution reduction, and (ii) a smoking prevention intervention. Second, for each intervention, we construct a hypothetical matched-pair randomized experiment, aiming at satisfying the "unconfoundedness" assumption (see Methods section). Third, we analyze the "unconfounded"/"as-if randomized" data subset with randomization-based inference to test sharp null hypotheses of no effect of the interventions for each unit at different taxonomic levels of the microbial ASV data. The results presented subsequently correspond to the third stage of the framework. Fourth, causal conclusions are developed in the Discussion section. Following the American Statistical Association statement [82, 94], we avoid searching for "statistically significant" results with a dichotomous approach. To give structure to our results reporting, we reject the sharp null hypotheses of no effect of an environmental intervention when the p-value is lower or equal to 0.1 or, when computed, when the adjusted p-value is lower or equal to 0.2. We are more tolerant with adjusted p-values because multiple comparison adjustments are conservative and our study is exploring a nearly untapped field. Nonetheless, we highly recommend to the readers interested in our research questions or result replication to examine all reported p-values in Figs and Tables, because higher p-values do not mean that an effect is improbable, absent, false, or unimportant [82].

## 3.1 Characteristics of study population

Our study is based on data from the KORA FF4 study cohort [17]. Because we performed a design stage before analyzing the data we have two study populations, one per hypothetical experiment, which are subsets of the entire cohort (see Design stage in the Methods section). In the air pollution reduction experiment, we analyze 99 matched pairs of subjects living in highly ($PM_{2.5} \geq 13.0\ \mu g/m^3$) and less ($PM_{2.5} \leq 10.3\ \mu g/m^3$) polluted areas with similar background characteristics distributions (Table 4 and Figs B-D and Fig H in S1 Text). The

**Table 4. Baseline characteristics of the study population in the air pollution reduction (left table) and smoking prevention experiments (right table).** Continuous variables: mean and standard deviation (St. d.). Categorical variables: number of samples per category (N) and proportion of category (%).

| | | Air pollution (PM$_{2.5}$) | | | | Smoking | | | |
|---|---|---|---|---|---|---|---|---|---|
| | | $\geq 13.0 \, \mu g/m^3$ | | $\leq 10.3 \, \mu g/m^3$ | | Smoker | | Never-Smoker | |
| | | Mean | St. d. | Mean | St. d. | Mean | St. d. | Mean | St. d. |
| Age | | 60.6 | 12.4 | 60.3 | 12.4 | 54.2 | 9.4 | 54.4 | 9.6 |
| Body Mass Index | | 27.0 | 4.3 | 27.0 | 3.8 | 26.7 | 4.4 | 26.7 | 4.2 |
| Alcohol intake (g/day) | | 11.3 | 14.1 | 11.5 | 13.9 | 13.0 | 15.6 | 11.6 | 14.3 |
| Years of education | | 11.9 | 2.6 | 11.7 | 2.8 | 11.7 | 2.3 | 11.8 | 2.2 |
| | | N | % | N | % | N | % | N | % |
| Sex | F | 41 | 20.7 | 41 | 20.7 | 130 | 24.0 | 130 | 24.0 |
| | M | 58 | 29.3 | 58 | 29.3 | 141 | 26.0 | 141 | 26.0 |
| Smoking | Ex-S. | 27 | 13.6 | 27 | 13.6 | - | - | - | - |
| | Never-S. | 62 | 31.3 | 62 | 31.3 | - | - | - | - |
| | Smoker | 10 | 5.1 | 10 | 5.1 | - | - | - | - |
| Diabetes | No | 95 | 48.0 | 95 | 48.0 | 264 | 48.7 | 264 | 48.7 |
| | Yes | 4 | 2.0 | 4 | 2.0 | 7 | 1.3 | 7 | 1.3 |
| Phys. Activity | No | 36 | 18.2 | 36 | 18.2 | 125 | 23.1 | 125 | 23.1 |
| | Yes | 63 | 31.8 | 63 | 31.8 | 146 | 26.9 | 146 | 26.9 |

thresholds for the air pollution experiment intervention are based on $90^{th}$ and $10^{th}$ percentiles of the PM$_{2.5}$ distribution. We focus on the PM$_{2.5}$ pollutant, originating mainly from traffic emissions and fossil fuel combustion, for its known penetrating effects into the lung and potential implication for the gut microbiome [27]. In the smoking prevention experiment, we analyze 271 matched pairs of smokers and never-smokers (with background characteristics distributions presented in Table 4 and Figs E-G and Fig I in S1 Text). A total of 45 units are included in the balanced data subset of both hypothetical experiments.

### 3.2 Microbial diversity analysis

A common first step in microbiome data analysis is estimating and assessing microbial diversity. We begin by investigating the potentially causal effects of the interventions on within-subject diversity ($\alpha$−diversity) and between-subject variation ($\beta$−diversity), respectively.

**3.2.1 Within-subject diversity.** Gut bacterial richness and Shannon diversity were estimated on the ASV level with the breakaway [83] and DivNet [55] method, respectively. Comparisons of the distributions of these estimated variables between subject under the intervention vs. not in both hypothetical experiments are shown by boxplots in Fig 2. The small approximate Fisherian p-values based on 10,000 permutations of the intervention assignment give us ground for rejecting the null hypotheses of no effect of an air pollution reduction (p-value$_{ap,richness} \approx$ 0.0008, p-value$_{ap,\alpha-div.} \approx$ 0.0388) and smoking prevention (p-value$_{s,richness} \approx$ 0.1518, p-value$_{s,\alpha-div.} \approx$ 0.0497) on the diversity of the human gut microbiome. On average, lower diversity was observed in the subjects living in polluted areas or smokers compared to participants living in less polluted areas or non-smokers. This diversity difference motivates the more in-depth analyses of the gut microbiome composition presented subsequently.

**3.2.2 Between-subject variation.** To estimate $\beta$-diversity indices, we calculated UniFrac, Aitchison, Jaccard, and Gower dissimilarities between all possible pairs of subjects. The results are shown in Table 5. To alleviate the problem of choosing the best dissimilarity metric for $\beta$−diversity estimation, we follow the Microbiome Regression-based Kernel Association Test

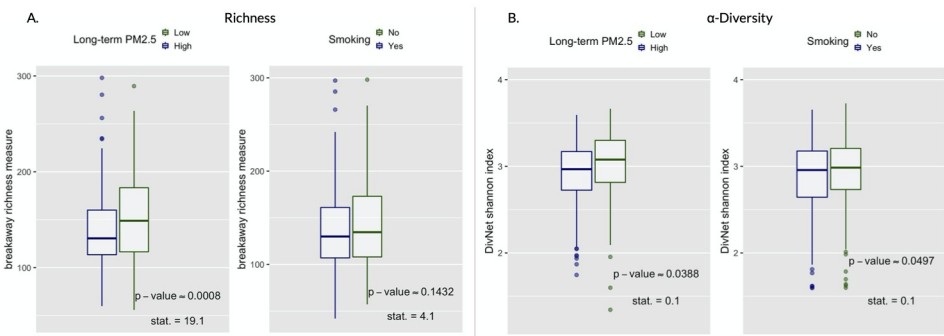

**Fig 2. Richness and *α*-diversity.** Boxplots (with median), values of the test-statistics from the `betta` regression [54], and one-sided randomization-based p-values for 10,000 permutations of the intervention assignment following a matched-pair design. (A) Boxplots of the richness. (B) Boxplots of the *α*-diversity.

(`MiRKAT`) of Zhao et al. [84] suggesting to compute several metrics and then adjust for multiple comparisons. In both experiments, we reject the sharp null hypotheses of no effect of the intervention on between-subject variation.

### 3.3 Microbial compositions analysis

We next investigated whether shifts in microbial compositions as a whole or differences in specific microbial taxa were observable in the hypothetical experiments. We illustrate this by designing and analyzing sharp null hypotheses for global compositional means and differential genus abundances.

**3.3.1 Compositional mean differences.**   Testing whether two study groups have the same microbiome composition can be viewed as a two-sample testing problem for high-dimensional compositional mean equivalence. We tested sharp null hypotheses using a test statistic developed particularly for that purpose by Cao et al. [56]. Table 6 summarizes the results for each taxonomic level. We reject the sharp null hypotheses of gut microbiome composition equivalence for the air pollution reduction and smoking prevention experiments. In both experiments, p-values are higher at the ASV level than at higher taxonomy levels.

**3.3.2 Differential taxon abundances.**   For compositional microbiome data, identifying sets of potentially "differentially abundant taxa" relates to testing sharp null hypotheses of no difference in abundance of individual taxa with respect to a reference set. We conducted such an analysis on the genus level for all genera present in at least 5% of the samples. This prevalence threshold was guided by the amount of information preserved when performing filtering, i.e., microbial abundance and the number of taxa observed per sample (see Figs N-Q in S1 Text). We applied the Differential abundance testing for compositional data (DACOMP) approach [57] and used two-sided tests since we lack prior knowledge on the direction of the

**Table 5. *β*-diversity.** Microbiome Regression-based Kernel Association Test (`MiRKAT`), unadjusted and adjusted one-sided randomization-based p-values for 10,000 permutations of the intervention assignment following a matched-pair design.

| | Air pollution | | | Smoking | | |
|---|---|---|---|---|---|---|
| *distance* | test-statistic | p-value | p-value$_{adj}$ | test-statistic | p-value | p-value$_{adj}$ |
| UniFrac | 12.1 | 0.0199 | 0.0506 | 61.5 | 0.0024 | 0.0070 |
| Aitchison | 82596.0 | 0.1096 | 0.2466 | 356921.5 | 0.0001 | 0.0003 |
| Jaccard | 19.4 | 0.0884 | 0.2043 | 84.5 | 0.0001 | 0.0003 |
| Gower | 0.2 | 0.0089 | 0.0250 | 0.1 | 0.0485 | 0.1204 |

**Table 6. Compositional equivalence test.** Test statistic for high-dimensional data suggested by [56] and one-sided randomization-based p-values for 10,000 permutations of the intervention assignment following a matched-pair design.

| | | ASV | Species | Genus | Family | Order | Class | Phylum |
|---|---|---|---|---|---|---|---|---|
| **Air Pollution** | nb. of taxa (p) | 4,370 | 414 | 252 | 74 | 44 | 29 | 15 |
| | test statistic | 12.8 | 12.9 | 11.9 | 8.8 | 8.4 | 8.4 | 8.1 |
| | p-value | 0.1451 | 0.0722 | 0.0733 | 0.1521 | 0.1161 | 0.1021 | 0.0591 |
| **Smoking** | nb. of taxa (p) | 7,409 | 479 | 271 | 81 | 48 | 31 | 16 |
| | test statistic | 13.0 | 14.5 | 13.3 | 11.6 | 8.6 | 9.4 | 10.4 |
| | p-value | 0.1607 | 0.0302 | 0.0384 | 0.0279 | 0.0859 | 0.0440 | 0.0135 |

abundance changes. Fig 3 highlights the key DACOMP results for both experiments. In the air pollution reduction experiment, we reject the sharp null hypothesis of no differential abundance only for the *Marvinbryantia* genus (p-value$_{adj.}$ = 0.0120) (see Table B in S1 Text). We also reject the sharp null hypothesis of no effect of smoking prevention for eleven genera (see Fig 3 and Table C in S1 Text). Five belong to the *Ruminococcaceae* family: *Ruminococcaceae-UCG-002*, *Ruminococcaceae-UCG-003*, *Ruminococcaceae-UCG-005*, *Ruminococcus-1*, and *Ruminococcaceae-NK4A214-group*, three to the *Lachnospiraceae* family: *Lachnospira*, *Lachnospiraceae-NK4A136-group*, and *Coprococcus-1*, one to the *Christensenellaceae* family: *Christensenellaceae-R-7-group*, and two to the *Mollicutes* class, which belong to the *NB1-n* and *Mollicutes-RF9* order.

### 3.4 Microbial network analysis

To gain insights into changes in the organizational structure of the underlying microbial gut ecosystem, we next calculated sparse genus-genus association networks for each exposure level and hypothetical experiment and highlight the results of our randomization-based differential association testing.

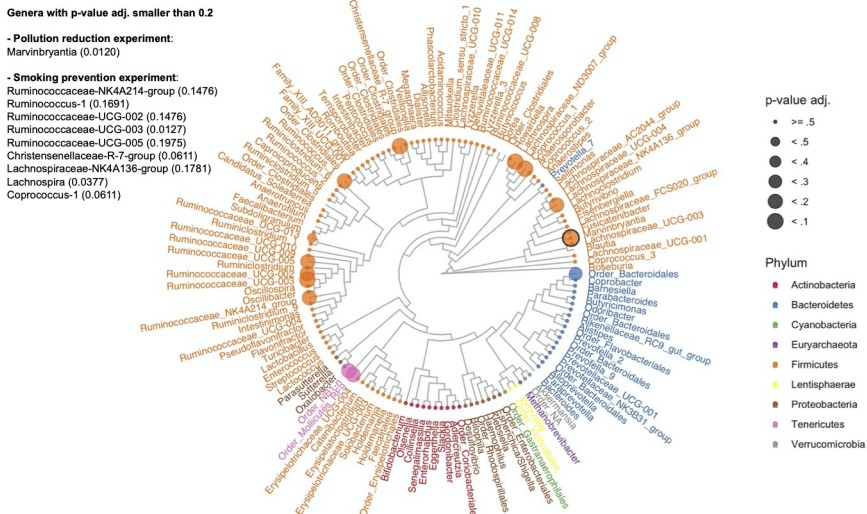

**Fig 3. Differential abundance.** For each genus, adjusted two-sided randomization-based p-values for 10,000 permutations of the smoking prevention intervention assignment following a matched-pair design. Genera with no tip point belong to the set of reference taxa. Black circled tip point: differentially abundant genus (*Marvinbryantia*) in the air pollution reduction experiment.

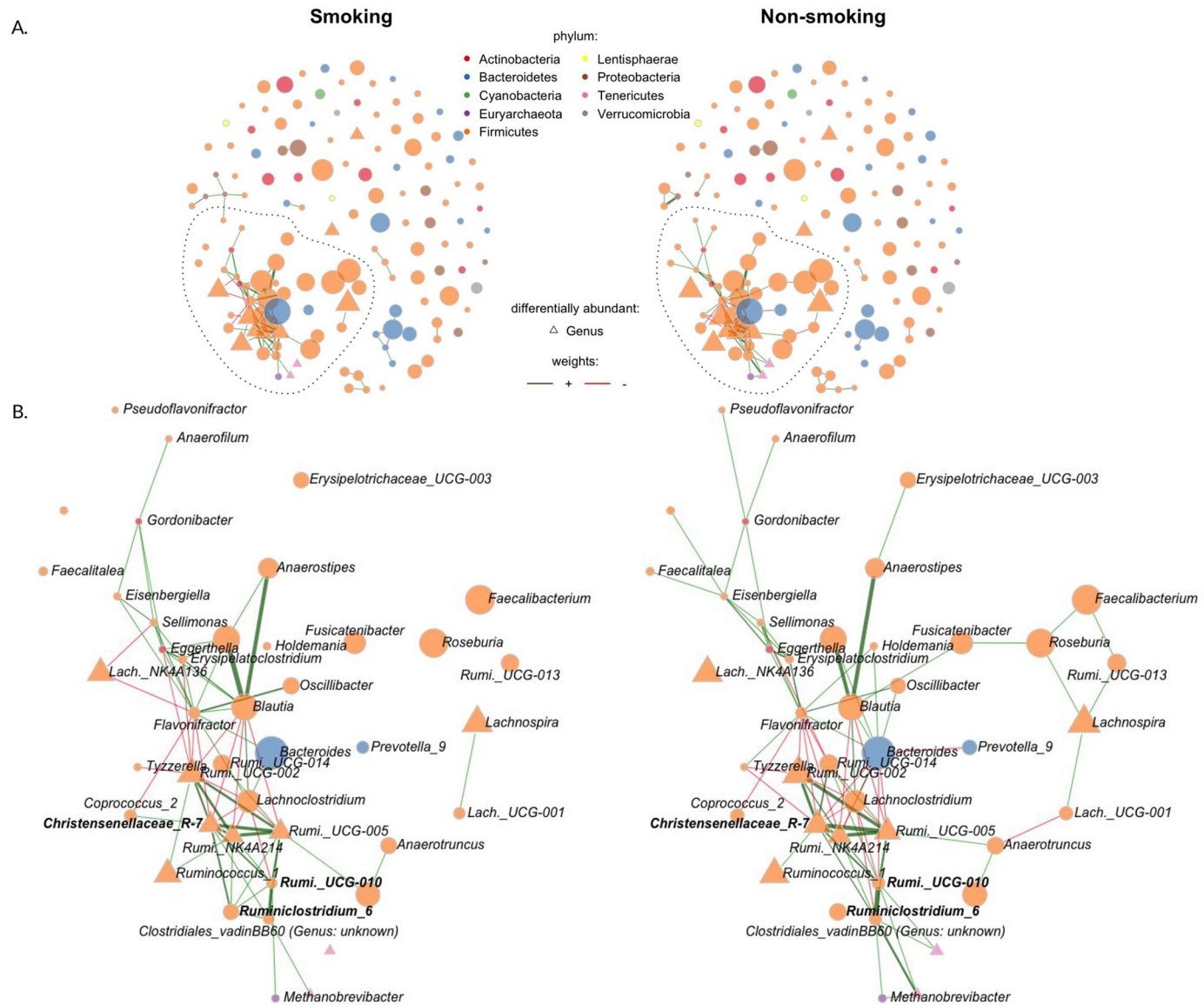

**Fig 4. Genus-genus associations of smokers and never-smokers (n = 271, p = 140).** (A) Visualization of the genus-genus partial correlations estimated with the SPIEC-EASI method. Edges thickness is proportional to partial correlation, and color to sign: red: negative partial correlation, green: positive partial correlation. Node size is proportional to the centered log ratio of the genus abundances, and color is according to phyla. Triangle shaped nodes are differentially abundant (see Fig 3). (B) Zoom in largest connected component and differential associations (bold genera).

**3.4.1 Genus-genus association networks.** We used the Sparse InversE Covariance estimation for Ecological ASsociation Inference (SPIEC-EASI) framework [58] to infer genus-genus associations in our two hypothetical experiments. We used the glasso mode of SPIEC-EASI with default parameters (see Methods for details). Fig 4A shows the overall structure of the learned sparse association networks for the smoking prevention experiment (smokers (left panel) and non-smokers (right panel), respectively). Each network possesses a single large connected component consisting of 30–40 mostly *Firmicutes* genera (highlighted area in Fig 4A). These connected components also included the majority of the previously identified potentially differentially abundant genera, including *Ruminococcaceae (UCG-005, UCG-002)*,

*Ruminococcus-1*, and *Christensenellaceae-R-7-group* (see Fig 4B for a detailed view of the connectivity pattern). The genus-genus associations networks derived from the air pollution reduction experiment showed similar overall topological features containing one large connected component of 60 genera, including *Ruminococcaceae (UCG-005, UCG-003, UCG-002)* and *Christensenellaceae-R-7-group* among others (see also Fig R in S1 Text).

**3.4.2 Differential genus-genus associations.** To identify potentially differential network associations in the intervention experiments, we coupled the SPIEC-EASI network estimation procedure with permutations of the intervention assignment, available in the `NetCoMi` R package [59] (see also Methods for details). For each hypothetical experiment, we list the five genus-genus associations with smallest adjusted two-sided randomization-based p-values in Table 7 and highlight these associations in Fig 4B. In the air pollution reduction experiment, we reject the sharp null hypothesis of no differential association for two edges: the *Succinivibrio/Slackia* edge (p-value$_{adj.} \approx 0.0661$), and the *Ruminiclostridium/Cloacibacillus* edge (p-value$_{adj.} \approx 0.1063$) (see Table 7 and Fig R in S1 Text).

In the smoking prevention experiment, we also reject the sharp null hypothesis of no differential association for two edges: the *Ruminiclostridium-6/Ruminococcaceae-UCG-010* edge (p-value$_{adj.} \approx 0.1585$), and the *Ruminiclostridium-6/Christensenellaceae-R-7-group* edge (p-value$_{adj.} \approx 0.1585$) (see Table 7). The genera that participate in these potentially differential associations are also highlighted in Fig 4B.

## 3.5 Exploring associations between genera and lipid metabolites

The gut microbiome is a substantial driver of circulating lipid levels, and prior work has shown [65, 95, 96] that the relative abundance of several microbial families, including *Christensenellaceae*, *Ruminococcaceae*, and the Tenericutes phylum were negatively correlated with triglyceride and positively associated with high-density lipoproteins (HDL) cholesterol. Since our analysis identified a small interconnected group of genera, including *Christensenellaceae* and *Ruminococcaceae*, for whom we rejected the no differential abundance hypothesis, we performed an exploratory data analysis to investigate taxa-serum lipid measurements associations. Four lipids were measured in blood serum samples of our study population from the KORA cohort: total, HDL, and LDL, cholesterol, as well as triglyceride levels. Fig 5A shows the correlation between these lipids and the genera we discovered in our hypothetical experiments. Tendencies similar to those reported in previous studies can be observed in our data.

For instance, in the smoking prevention dataset, we observed a positive correlation of Christensenellaceae R-7-group and Ruminococcaceae (UCG-005) genus abundances (under centered log-ratio transformation) with HDL cholesterol and negative correlation with triglyceride levels, respectively (see Fig 5B). Similar correlation patterns were also found for the other genera for whom we rejected the no differential abundance hypothesis (see second and forth column in Fig 5A). Our findings were also in line with recently reported correlation results in Vojinovic et al. [65] using the Dutch LifeLines-DEEP cohort [13] and the Rotterdam Study [14].

## 3.6 Sensitivity analysis

To assess whether the pair-matching strategy chosen for the design stage influenced the conclusions of this study, we conducted a sensitivity analysis (see Sensitivity Analysis section in S1 Text). For that, we implemented the more commonly-used propensity score matching algorithm [97] and obtained matched samples of: 1) 158 participants living in low $PM_{2.5}$ areas and 158 participants living in higher $PM_{2.5}$ areas, and 2) 290 smokers and 290 never smokers (see Table D and Figs T-Y in S1 Text for the balance diagnostics). For both hypothetical

**Table 7. Differential associations of genera.** Smallest five adjusted two-sided randomization-based p-values for 10,000 permutations of the intervention assignment following a matched-pair design.

| Air pollution | |
|---|---|
| Genus-genus associations (-: disappearance after intervention) | p-value$_{adj}$ |
| *Succinivibrio/Slackia* (-) | 0.0661 |
| *Ruminiclostridium/Cloacibacillus* (-) | 0.1063 |
| *Cloacibacillus/Lachnospiraceae-FCS020-group* | 0.2795 |
| *Megasphaera/Alistipes* | 0.4147 |
| *Bacteroidales* (Genus: unknown)/*Prevotella-2* | 0.4753 |
| **Smoking** | |
| Genus-genus associations (-: disappearance after intervention) | p-value$_{adj}$ |
| *Christensenellaceae-R-7/Ruminiclostridium-6* (-) | 0.1585 |
| *Ruminococcaceae-UCG-010/Ruminiclostridium-6* (-) | 0.1585 |
| *Ruminococcaceae-UCG-014/Flavonifractor* | 0.2031 |
| *Clostridiales-vadinBB60/Ruminiclostridium-6* | 0.2376 |
| *Ruminococcaceae-UCG-013/Faecalibacterium* | 0.2492 |

randomized experiments, using propensity score matching at the design stage results in analyzing more matched samples. The microbial diversity analyses lead to the same conclusion for both experiments despite different design stages (see Fig Z and Tables E-F in S1 Text). Overall, we also observe small approximate Fisherian p-values after performing the propensity score matching, in the same way we observe small approximate Fisherian p-values with our pair-matching strategy. The test statistics have the same direction and magnitude. For the air pollution reduction experiment, the adjusted p-values are higher when performing propensity score matching when checking for differential abundances, i.e., we cannot reject the sharp null hypothesis of no differential abundance for the *Marvinbryantia* genus. For the smoking prevention experiment, we can reject the sharp null of no differential abundance for the same taxa and additional ones when performing propensity score matching compared to pair-matching (see Table C and Table G in S1 Text).

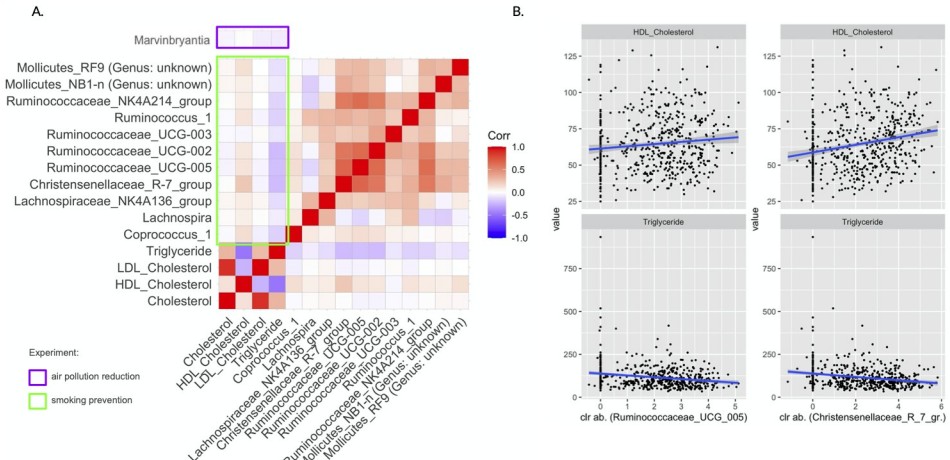

**Fig 5. Lipid metabolites exploration.** (A) Lipid metabolites correlation with selected genera from the smoking prevention experiment (green). (B) Scatterplots of high-density lipoprotein (HDL) cholesterol and triglycerides vs. centered log-ratio transformed relative abundances of the genera *Ruminococcaceae-UCG-005* and *Christensenellaceae-R-7-group*.

## 4 Discussion

We first discuss the results presented above, then elaborate on the statistical framework we used for our analyses, and suggest statistical and epidemiological extensions of our work.

In the air pollution ($PM_{2.5}$) reduction hypothetical experiment, we reject the sharp null hypotheses of no richness, no $\alpha$-diversity, no $\beta$-diversity, and no high-dimensional mean differences. We also reject the no differential abundance hypothesis for the Marvinbryantia genus, and the no differential association hypothesis between: the *Succinivibrio* and *Slackia* genera, as well as the *Ruminiclostridium* and *Cloacibacillus* genera. Experiments exposing mice to $PM_{2.5}$ resulted in mixed findings concerning difference in microbial richness and diversity. This might be due to regional differences in the chemical composition of $PM_{2.5}$ as well as differences in the duration of exposure [29]. Thus far, only one human study estimated associations between $PM_{2.5}$ exposure and the gut microbiome, and investigated the pathway of diabetes induction associated with PM exposure [28]. One of their key findings was that $PM_{2.5}$ exposure reduced $\alpha$-diversity (measured by Chao1 and Shannon indices), which is consistent with our observations.

In the smoking prevention hypothetical experiment, we rejected the sharp null hypotheses of no richness, no $\alpha$-diversity, no $\beta$-diversity, and no high-dimensional mean differences. We also rejected the no differential abundance hypothesis for eleven genera (five of the *Ruminococcaceae* family, three of the *Lachnospiraceae* family, one of the *Christensenellaceae* family, and two of the *Mollicutes class*), and the no differential association hypothesis between the *Ruminiclostridium-6* and *Ruminococcaceae-UCG-010* genera, and between the *Ruminiclostridium-6* and *Christensenellaceae R-7-group* genera. Interestingly, the associations of *Ruminococcaceae-UCG-010* and *Christensenellaceae R-7-group* with *Ruminiclostridium-6* were also found to be worth further scrutiny. Their positive associations in the genus-genus network of smokers was absent in the genus-genus network of the never-smokers. The one study comparing the gut microbiome of smokers ($n = 203$) and never-smokers ($n = 288$) with similar sample size has a men-only study population [44]. They did not find any differences in $\alpha$-diversity (measured with the Shannon index), whereas we conclude that $\alpha$-diversity analyses are worth further scrutiny. Lee et al.'s PERMANOVA analyses for $\beta$-diversity differences, measured with Jaccard and weighted UniFrac distances, suggested differences. We reject the sharp null hypothesis at the between-subject differences analysis level. In their analysis of bacterial taxa on the phylum level, smokers had an increased proportion of Bacteroidetes with decreased Firmicutes and Proteobacteria compared with never-smokers. When we compare these phyla, we do not observe the same differences (see Fig S in S1 Text). Also, our compositional difference analyses do not result in the same set of differentially abundant genera that were reported by Lee et al. [44]. These conflicting findings could be due to the fact that their study was done on Korean men only. Nonetheless, it shows that there is a lack of knowledge on the effects of smoking on the human gut microbiome and that additional scientific investigations are necessary to make causal conclusions.

Throughout the extensive statistical analyses presented in this paper, we have tested sharp null hypotheses of no effect of an intervention on a wide range of gut microbiome outcomes, ranging from high-level microbial diversity estimates to differential genus-genus associations. To do so, we have performed randomization-based inference based on 10,000 permutations. This mode of inference has been motivated by two reasons: (i) it is difficult to postulate a joint model for the potential outcomes, and thereby provide an estimate of (and uncertainty around) a causal estimand, and (ii) it has been shown that using the actual randomization procedure that led to the observed data helps to report valid Fisher-exact p-values as opposed to p-values relying on approximating null randomization distributions [46]. As an example, in

our mean difference analyses, we found some differences between the null randomization distribution of the test statistic when approximated by permuting the intervention assignment vector and when drawn from the approximating asymptotic distribution (see Figs J-K in S1 Text). A natural extension of this study would be to use a Neymanian or Bayesian mode of inference to tackle the same research questions. There, simulations should support evidence whether the approach can indeed recover the then estimated causal effects. Simulating microbiome data requires effort so that the common properties, such as compositionality and zero-inflation, can be preserved, but re-sampling approaches [98] and generative models [99] have been developed to achieve this end.

An important component of our randomization-based procedure is that the permutations of the intervention assignment vector conserves the matched-pair design of the hypothetical randomized experiment. This strategy has been advocated by Rubin [100] in the context of randomized trials, and more recently by Bind and Rubin [46] in the context of hypothetical randomized experiments, because assumptions on the underlying distribution of the data are not required. Only few R packages were built to perform randomization-based inference while conserving the design of the intervention assignment. Therefore, for every analysis in our study, we imported a matrix of 10,000 unique randomized intervention assignments to calculate our p-values (see https://github.com/AliceSommer/Causal_Microbiome_Tutorial for a reproducible example on the American Gut Data [16, 101]). Nonetheless, the DACOMP and NetCoMi R packages provide flexible functions enabling the calculation of randomization-based p-values for our study design to test sharp null hypotheses of no difference in taxa abundance and associations, respectively. We advocate for more development of such user-friendly software functions permitting flexibility and accountability of the design stage of observational studies. P-value adjustments for multiple comparison also follow a fully randomization-based procedure, while preserving the design of the experiment. The method has proven to be more powerful while maintaining the family-wise error rate [91].

Notice that when presenting our results, we never accepted alternative hypotheses but only rejected sharp nulls when unadjusted and adjusted p-values were small, i.e., indicating the hypotheses warrants further scrutiny [82]. In the field of microbiome data analysis, the terms differential abundance and associations are frequently used. Researchers report "differentially abundant" and "differentially associated" sets of taxa after testing sharp null hypotheses of no effect of an intervention. This terminology implicitly implies acceptance of the alternative hypotheses. However, when testing sharp null hypotheses we assess the amount of evidence against them in the observed data, which does not prove the alternative hypothesis to be true.

During the design stage, the outcome variable was ignored and only pre-exposure covariates were considered. The chosen balanced data is a sub-sample of units that can be used to estimate the effects of an intervention. Omitting the outcome data until the analysis avoids "model cherry-picking", because the effect of the intervention is estimated once, after a successful design stage. Nonetheless, at the design stage, we can only consider the observed pre-exposure variables but the assignment mechanism could depend on unobserved pre-exposure variables. In gut microbiome studies, diet is often an unobserved confounder. For example, in this study, dietary intake data was collected for only 1,469/2,033 (i.e., 72%) participants. We verified balance in dietary intake for our balanced data subset (see Figs H-I in S1 Text). Even though we made sure that the observed potential confounding covariates are fairly balanced, there could still be imbalances in other unobserved background covariates, which could have an effect on our results. In such cases, Rosenbaum [102] has recommended to consider sensitivity analyses of how the Fisher-exact p-value would change, had the intervention assignment been plausibly different, see also Bind and Rubin [46]. Subject-matter knowledge on the probability of the binary exposure (i.e., smoking or air pollution) given the observed and unobserved

background covariates should guide the plausible range of "sensitivity" p-values and the reason why they could deviate from the p-value calculated based on the assumed hypothetical intervention assignment. This idea provides material for an extension of the framework presented in this study.

The framework suggested in this paper facilitates a more transparent interpretation of results than standard approaches directly modeling the observed outcome. First, interpretation is only valid within the range of the background covariates of the study population in the respective hypothetical experiment (see their detailed characteristics in Table 4 and Figs B-I in S1 Text). The data do not provide direct information for the "unmatched" units. In addition to our pair-matching strategy, we conducted a sensitivity analysis using a propensity score matching algorithm at the design stage, which led to more matched pairs, and thus a broader range of background covariates values (see Table D in S1 Text). Both matching algorithms do not lead to conflicting results in the smoking prevention experiments. In the air pollution reduction experiment, only the differential abundance analysis does not lead to the same overall conclusion. At this stage, the researcher can decide between a larger number of units or more similar groups of units to compare. When designing our hypothetical experiment, we chose a pair-matching strategy, because it creates similar pairs of participants based on subject-matter knowledge. For example, the number of females and males in the intervention and control groups is identical after pair-matching, whereas with propensity score matching, these numbers slightly differ (see Table 4 and Table D in S1 Text). Note that the matching algorithm considerations should be *a priori* specified before any statistical analysis is performed. Ideally, the design stage should be conducted by a statistician who is not involved in the subsequent statistical analysis stage. Second, the assumed assignment mechanism and underlying assumptions have to be clearly stated to obtain meaningful p-values. Standard approaches usually make strong assumptions (e.g., linearity), whose discussions are often neglected. Modeling the observed data and solely adjusting for confounders by including them in a regression, without a design stage, can be unreliable, especially when the pre-exposure covariates distributions of the control and intervention units are not similar. For instance, Cochran and Rubin [47], Heckman et al. [103], and Rubin [104] have shown that regression models can estimate biased treatment effects when the true relationship between the covariates and the outcome is not modeled accurately. Dehejia and Wahba have also shown that standard nonexperimental estimators such as regression are sensitive to the specification used in the regression [105]. This is another reason why we opted for an inference method that does not rely on parametric assumptions.

In contrast to other studies interested in the effect of air pollution exposures on health outcomes, this study does not provide any estimation of an exposure-response curve. Instead, we examine the effect of interventions and provide results that can directly contribute to policy recommendations. Until now, relationships between inhaled environmental exposures and the human gut microbiome were not examined with causal inference methods, so a first step to make advances in the field is to test, whether air pollution and smoking have no effect on the units of our study. If so, a potential next step would be to work with a dataset adequate for balancing covariates along different doses of the exposure such as suggested in [106] and estimate a causal dose-response in order to protect populations at risk.

In the smoking prevention experiment, the subset of genera retained at the differential abundance analysis step was linked to the serum markers triglycerides and high-density lipoprotein in previous studies [65, 95, 96]. In our data, we observe correlations between these genera and metabolites in the same direction than previously found by Vojinovic [65] (see Fig 5). Serum triglycerides and high-density lipoprotein play a role in metabolic syndrome, and associations between smoking and metabolic syndrome have also been found previously [107]. Therefore, we suggest further investigation on the pathway of cigarette smoke impacting the

gut, which in turn has effects on circulating metabolites (and metabolic syndrome). A logical next step would be to apply our framework to other cohorts with similar amplicon data pre-processing and available pre-exposure covariates such as the Dutch LifeLines-DEEP [13] and Rotterdam Studies [14], and observe whether our results replicate.

## Supporting information

**S1 Text. Fig A**: Gut microbiome data description. Number of observed ASV per sample (top left), sequencing depth per sample (top right), number of sequences per ASV (bottom left), number of zero count per ASV (bottom right). **Fig B**: Empirical distributions of the matched covariates among the subjects under the intervention vs. not in the original (left panel) and the balanced (right panel) data for the air pollution reduction hypothetical experiment. **Fig C**: Empirical distributions of the disease covariates among the subjects under the intervention vs. not in the original (left panel) and the balanced (right panel) data for the air pollution reduction hypothetical experiment. **Fig D**: Empirical distributions of the medication covariates among the subjects under the intervention vs. not in the original (left panel) and the balanced (right panel) data for the air pollution reduction hypothetical experiment. **Fig E**: Empirical distributions of the matched covariates among the subjects under the intervention vs. not in the original (left panel) and the balanced (right panel) data for the smoking prevention hypothetical experiment. **Fig F**: Empirical distributions of the diseases covariates among the subjects under the intervention vs. not in the original (left panel) and the balanced (right panel) data for the smoking prevention hypothetical experiment. **Fig G**: Empirical distributions of the medication covariates among the subjects under the intervention vs. not in the original (left panel) and the balanced (right panel) data for the smoking prevention hypothetical experiment. **Fig H**: Empirical distributions of the nutrition covariates among the subjects under the intervention vs. not in the balanced data for the air pollution reduction hypothetical experiment. **Fig I**: Empirical distributions of the nutrition covariates among the subjects under the intervention vs. not in the balanced data for the smoking prevention hypothetical experiment. **Fig J**: Permutation-based (grey) and asymptotic (blue) null randomization distributions for the air pollution reduction hypothetical experiment. **Fig K**: Permutation-based (grey) and asymptotic (blue) null randomization distributions for the smoking prevention hypothetical experiment. **Fig L**: Reference set selection in the air pollution reduction experiment. A taxa enters the set $R = (r_1, \ldots, r_F)$ if it has low variance ($< 2$) and high prevalence ($> 90\%$). For the analyses at the ASV level, we chose the variance to be $< 3$ and the prevalence to be $> 40\%$ as thresholds in order the have at least one reference per subject. **Fig M**: Reference set selection in the smoking prevention experiment. A taxa enters the set $R = (r_1, \ldots, r_F)$ if it has low variance ($< 2$) and high prevalence ($> 90\%$). For the analyses at the ASV level, we chose the variance to be $< 3$ and the prevalence to be $> 40\%$ as thresholds in order the have at least one reference per subject. **Fig N**: Distribution of number of ASVs per sample when data is filtered at different ASV prevalence thresholds (0%, 5%, 10%, 15%) in the air pollution reduction experiment. Red value: minimum observed ASVs per sample. **Fig O**: Distribution of the total ASV counts per sample when data is filtered at different ASV prevalence thresholds (0%, 5%, 10%, 15%) in the air pollution reduction experiment. Red value: minimum ASV counts per sample. **Fig P**: Distribution of number of ASVs per sample when data is filtered at different ASV prevalence thresholds (0%, 5%, 10%, 15%) in the smoking prevention reduction experiment. Red value: minimum observed ASVs per sample. **Fig Q**: Distribution of the total ASV counts per sample when data is filtered at different ASV prevalence thresholds (0%, 5%, 10%, 15%) in the smoking prevention experiment. Red value: minimum ASV counts per sample. **Fig R**: Genus-genus associations for subject under the air pollution reduction experiment vs. not (n = 99, p = 149).

(A) Visualization of the between genera partial correlations estimated with the SPIEC-EASI method. Edges thickness is proportional to partial correlation, and color to direction: red: negative partial correlation, green: positive partial correlation. Node size is proportional to the centered log ratio of the genus abundances, and color is according to phyla. Triangle shaped nodes are differentially abundant (see Fig 3). (B) Zoom in largest connected component and differential associations (bold genera). **Fig S**: Phyla comparison. **Fig T**: Sensitivity analysis—Empirical distributions of the matched covariates among the subjects under the intervention vs. not in the original (left panel) and the balanced (right panel) data for the air pollution reduction hypothetical experiment. **Fig U**: Sensitivity analysis—Empirical distributions of the diseases covariates among the subjects under the intervention vs. not in the original (left panel) and the balanced (right panel) data for the air pollution reduction hypothetical experiment. **Fig V**: Sensitivity analysis—Empirical distributions of the medication covariates among the subjects under the intervention vs. not in the original (left panel) and the balanced (right panel) data for the air pollution reduction hypothetical experiment. **Fig W**: Sensitivity analysis—Empirical distributions of the matched covariates among the subjects under the intervention vs. not in the original (left panel) and the balanced (right panel) data for the smoking prevention hypothetical experiment. **Fig X**: Sensitivity analysis—Empirical distributions of the diseases covariates among the subjects under the intervention vs. not in the original (left panel) and the balanced (right panel) data for the smoking prevention hypothetical experiment. **Fig Y**: Sensitivity analysis—Empirical distributions of the medication covariates among the subjects under the intervention vs. not in the original (left panel) and the balanced (right panel) data for the smoking prevention hypothetical experiment. **Fig Z**: Sensitivity analysis—Richness and $\alpha$-diversity. Boxplots (with median), values of the test-statistics from the `betta` regression, and one-sided randomization-based p-values for 10,000 permutations of the intervention assignment following a matched-pair design. **Table A**: Gut microbiome data description. Number of observed ASV per sample, sequencing depth per sample, number of sequences per ASV, number of zero count per ASV. **Table B**: Air pollutiion reduction experiment results. Differentially abundant taxa and adjusted Fisher p-values for 10,000 iterations at 5% prevalence filtering. Selected adjusted p-values $\leq 0.2$ (sign of abundance difference: y(1)—y(0)). **Table C**: Smoking prevention experiment results. Differentially abundant taxa and adjusted Fisher p-values for 10,000 iterations at 5% prevalence filtering. Selected adjusted p-values $\leq 0.2$ (sign of abundance difference: y(1)—y(0)). **Table D**: Sensitivity analysis—Baseline characteristics of the study population in the air pollution reduction (left table) and smoking prevention experiments (right table). Continuous variables: mean and standard deviation (St. d.). Categorical variables: number of samples per category (N) and proportion of category (%). **Table E**: Sensitivity analysis—$\beta$-diversity. Microbiome Regression-based Kernel Association Test (`MiRKAT`), unadjusted and adjusted one-sided randomization-based p-values for 10,000 permutations of the intervention assignment following a matched-pair design. **Table F**: Sensitivity analysis—Compositional equivalence test. Test statistic for high-dimensional data and one-sided randomization-based p-values for 10,000 permutations of the intervention assignment following a matched-pair design. **Table G**: Sensitivity analysis—Smoking prevention experiment results. Differentially abundant taxa and adjusted Fisher p-values for 10,000 iterations at 5% prevalence filtering. Selected adjusted p-values $\leq 0.2$ (sign of abundance difference: y(1)—y(0)).
(PDF)

## Acknowledgments

We thank all KORA participants and technical assistants without whose contributions this study could not have been realized. We also thank Stefanie Peschel and Viet Tran for testing

the code for the tutorial with the American Gut Data as well as Barak Brill for his support in the DACOMP implementation. The computations in this paper were run on the FASRC Odyssey cluster supported by the FAS Division of Science Research Computing Group at Harvard University.

## Author Contributions

**Conceptualization:** Alice J. Sommer, Annette Peters, Christian L. Müller, Marie-Abèle C. Bind.

**Funding acquisition:** Annette Peters, Marie-Abèle C. Bind.

**Methodology:** Alice J. Sommer, Christian L. Müller, Marie-Abèle C. Bind.

**Supervision:** Annette Peters, Christian L. Müller, Marie-Abèle C. Bind.

**Visualization:** Alice J. Sommer.

**Writing – original draft:** Alice J. Sommer.

**Writing – review & editing:** Alice J. Sommer, Annette Peters, Martina Rommel, Josef Cyrys, Harald Grallert, Dirk Haller, Christian L. Müller, Marie-Abèle C. Bind.

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
