## [Decision Letter · Decision Letter 0]

25 Oct 2021

Dear Dr. Mueller,

Thank you very much for submitting your manuscript "A randomization-based causal inference framework for uncovering environmental exposure effects on human gut microbiota" for consideration at PLOS Computational Biology.

As with all papers reviewed by the journal, your manuscript was reviewed by members of the editorial board and by several independent reviewers. In light of the reviews (below this email), we would like to invite the resubmission of a significantly-revised version that takes into account the reviewers' comments.

GUEST EDITOR'S COMMENTS:

Reviewer #2, who is overall very positive, wrote that "the strength of the paper lies in applying the Rubin Causal Model to the data sets to go beyond correlational and associational explorations", and this, I suppose, is a fair summary of your paper's main selling point. Nevertheless, the Reviwer considers the "organization of the paper [as] awkward".

In contrast to Reviewer #1, Reviewer #3 seems to doubt that you are really using a causal frame: "the framework is nothing but just matching". I do not know much about causal inference, myself, but I also had this thought when reading your paper: if normal inference of correlation cannot show direction of causality, how can a matching scheme like your overcome that? I suppose that the answer to this question is the core idea of Rubin's model, and hence, maybe you need add a more thorough introduction to Rubin's methodology, accessible to readers not yet familiar with Rubin's work, and make sure that it is well linked to your actually method and explains how your matching-based scheme allows to actually infer causation rather than only correlation.

Reviewer #1 does not seem to share Reviewer #3's and my doubts about how matching can recover causal relations (and this is why I suppose that my doubts are merely due to my lack of understaning of Rubin's work and how you apply it). Nevertheless, that reviewer has several questions on how you justify your selection of a matching scheme, and how it compares to other state-of-the-art approaches. The reviewer also is unconvinced that your method is actually able to recover causal relationships, and suggests to prove this with simulations.

Therefore, I would ask you to carefully address all the reviewers' concerns and especially improve the explanation on how the method actually achieves causal inference. This may require a reorganizing of the manuscript as suggested by Reveiwer #2.

----

We cannot make any decision about publication until we have seen the revised manuscript and your response to the reviewers' comments. Your revised manuscript is also likely to be sent to reviewers for further evaluation.

Sincerely,

Simon Anders

Guest Editor

PLOS Computational Biology

Kiran Patil

Deputy Editor

PLOS Computational Biology

GUEST EDITOR'S COMMENTS:

Reviewer #2, who is overall very positive, wrote that "the strength of the paper lies in applying the Rubin Causal Model to the data sets to go beyond correlational and associational explorations", and this, I suppose, is a fair summary of your paper's main selling point. Nevertheless, the Reviwer considers the "organization of the paper [as] awkward".

In contrast to Reviewer #1, Reviewer #3 seems to doubt that you are really using a causal frame: "the framework is nothing but just matching". I do not know much about causal inference, myself, but I also had this thought when reading your paper: if normal inference of correlation cannot show direction of causality, how can a matching scheme like your overcome that? I suppose that the answer to this question is the core idea of Rubin's model, and hence, maybe you need add a more thorough introduction to Rubin's methodology, accessible to readers not yet familiar with Rubin's work, and make sure that it is well linked to your actually method and explains how your matching-based scheme allows to actually infer causation rather than only correlation.

Reviewer #1 does not seem to share Reviewer #3's and my doubts about how matching can recover causal relations (and this is why I suppose that my doubts are merely due to my lack of understaning of Rubin's work and how you apply it). Nevertheless, that reviewer has several questions on how you justify your selection of a matching scheme, and how it compares to other state-of-the-art approaches. The reviewer also is unconvinced that your method is actually able to recover causal relationships, and suggests to prove this with simulations.

Therefore, I would ask you to carefully address all the reviewers' concerns and especially improve the explanation on how the method actually achieves causal inference. This may require a reorganizing of the manuscript as suggested by Reveiwer #2.

----

Reviewer's Responses to Questions

**Comments to the Authors:**

Reviewer #1: See attached.

Reviewer #2: see attached

Reviewer #3: The paper introduces a causal inference framework to investigate the treatment effect of environmental factors on the human microbiome. The core idea is to use matched pairs to balance the pre-exposure characteristics of participants and then to use randomization-based inference. The authors illustrated their framework on the German KORA cohort study, specifically, the effects of air pollution reduction and smoking prevention on the human gut microbiome. The paper is easy to follow and includes various analyses in the proposed framework. However, the framework is nothing but just matching, where sensitivity analysis is an essential part. The authors mentioned sensitivity analysis in the Discussion, but the reviewer doesn't think that is enough. Results for sensitivity analysis should be included, as well-known factors that are associated with the microbiome, such as diet and medication, were not used in matching. It is also not clear how their results differ from the analyses with covariate adjustments, which is commonly performed in microbiome studies. The authors mentioned the unreliability of regression with covariate adjustment in the Discussion, but matching is also not reliable if there are unmeasured covariates that have confounding effects. It would be very instructive if the authors could demonstrate that matching is more reliable than covariate adjustment in the microbiome study.

Minor:

Column titles in Table 1 are switched.

It seems rather subjective in determining thresholds for covariates in matching. How was the threshold for each covariate determined?

**Have the authors made all data and (if applicable) computational code underlying the findings in their manuscript fully available?**

Reviewer #1: Yes

Reviewer #2: Yes

Reviewer #3: Yes

PLOS authors have the option to publish the peer review history of their article (what does this mean?). If published, this will include your full peer review and any attached files.

Reviewer #1: No

Reviewer #2: No

Reviewer #3: No
---

## [Decision Letter · Decision Letter 1]

24 Feb 2022

Dear Dr. Mueller,

Thank you very much for submitting your manuscript "A randomization-based causal inference framework for uncovering environmental exposure effects on human gut microbiota" for consideration at PLOS Computational Biology.

As with all papers reviewed by the journal, your manuscript was reviewed by members of the editorial board and by several independent reviewers. In light of the reviews (below this email), we would like to invite the resubmission of a significantly-revised version that takes into account the reviewers' comments.

---8<---

Guest Editor's comments:

I would like to thank the authors for their revised submission and the reviewers for their detailled and insightful comments on it. I'm enjoying being guest editor for this paper as it prompted me to read up on Rubin causality, and I have learned a lot from working through paper and comments.

Reviewer #2 had much praise for the improved paper, and I also agree with them that the author's reorganization of the paper has improved readability greatly. Reviewers #1 and #3 still have a few major concerns. Overall, I tend to agree with reviewer #2 that the paper is now in a good shape, pending the authors addressing the numerous minor comments made by all three reviewers. I would ask you to go through them carefully and amend and clarify the text acordingly.

The authors have rightfully pointed out that by applying Rubin's causal model to high-dimensional data and complex hypotheses that intermingle the dimensions (such as diversity measures) they are the first to bridge two subjects so far not connected, and that this paper is therefore a first step and cannot be expected to address every aspect of the topic. In this light, I would say that some of the Reviewers' remaining major concern merit a paper in its own right and would not need to be addressed here, provided the necessity for such further research is mentioned in the discussion.

Specifically:

- Reviewer #1 is certainly right that PSM is the only workable solution if one has many covariates, but the present experiment has only few -- so we can leave this issue to whoever is the first to actually face a situation with many covariates. For the present case of few covariates, the authors consider it obvious that individual matching beats PSM, the Reviewer doubts that. Hence, this statement should be either elaborated or dropped.

- I do not quite agree with Reviewer #1 on the need for simulations to ensure that null distributions are as expected. A simulation cannot help us to assess whether matching properly remedies confounding. However, once we assume there to be no confounding, we are in the same situation as in an actual randomized study. The question whether the tests employed by the authors are appropriate for randomized studies or controlled experiments in metabolomics have been discussed thoroughly in the papers that introduced these tests to the field of metabolomics -- and hopefully checked by simulations there. So, citing existing literature seems sufficient to me here.

- Reviewer #3 critizises the lack of a quantitative analysis of sensitivity to confounding. I imagine that here the bigger issue is not residual confounding to the (very few) known covariates but the influence of unknown/unrecorded covariates. Hence, while ideally, the authors would add a quantitative sensitivity anaysis in the Reviewer's sense, a qualitative discussion of the risk of (a) insufficient compensation for the known confounders and (b) unaccounted confounders should be sufficient, too.

- Regarding Reviewer #2's second comment, on comparing matching on a subset with accounting and using all samples: This is a question of gaining optimal power, not of ensuring correctness of inference. As the present paper does not claim to be the final word on the topic, such ways of optimizing inferential power are arguably beyond its scope, so we can leave this discussion to some future work.

---8<--

We cannot make any decision about publication until we have seen the revised manuscript and your response to the reviewers' comments. Your revised manuscript is also likely to be sent to reviewers for further evaluation.

Sincerely,

Simon Anders

Guest Editor

PLOS Computational Biology

Kiran Patil

Deputy Editor

PLOS Computational Biology

I would like to thank the authors for their revised submission and the reviewers for their detailled and insightful comments on it. I'm enjoying being guest editor for this paper as it prompted me to read up on Rubin causality, and I have learned a lot from working through paper and comments.

Reviewer #2 had much praise for the improved paper, and I also agree with them that the author's reorganization of the paper has improved readability greatly. Reviewers #1 and #3 still have a few major concerns. Overall, I tend to agree with reviewer #2 that the paper is now in a good shape, pending the authors addressing the numerous minor comments made by all three reviewers. I would ask you to go through them carefully and amend and clarify the text acordingly.

The authors have rightfully pointed out that by applying Rubin's causal model to high-dimensional data and complex hypotheses that intermingle the dimensions (such as diversity measures) they are the first to bridge two subjects so far not connected, and that this paper is therefore a first step and cannot be expected to address every aspect of the topic. In this light, I would say that some of the Reviewers' remaining major concern merit a paper in its own right and would not need to be addressed here, provided the necessity for such further research is mentioned in the discussion.

Specifically:

- Reviewer #1 is certainly right that PSM is the only workable solution if one has many covariates, but the present experiment has only few -- so we can leave this issue to whoever is the first to actually face a situation with many covariates. For the present case of few covariates, the authors consider it obvious that individual matching beats PSM, the Reviewer doubts that. Hence, this statement should be either elaborated or dropped.

- I do not quite agree with Reviewer #1 on the need for simulations to ensure that null distributions are as expected. A simulation cannot help us to assess whether matching properly remedies confounding. However, once we assume there to be no confounding, we are in the same situation as in an actual randomized study. The question whether the tests employed by the authors are appropriate for randomized studies or controlled experiments in metabolomics have been discussed thoroughly in the papers that introduced these tests to the field of metabolomics -- and hopefully checked by simulations there. So, citing existing literature seems sufficient to me here.

- Reviewer #3 critizises the lack of a quantitative analysis of sensitivity to confounding. I imagine that here the bigger issue is not residual confounding to the (very few) known covariates but the influence of unknown/unrecorded covariates. Hence, while ideally, the authors would add a quantitative sensitivity anaysis in the Reviewer's sense, a qualitative discussion of the risk of (a) insufficient compensation for the known confounders and (b) unaccounted confounders should be sufficient, too.

- Regarding Reviewer #2's second comment, on comparing matching on a subset with accounting and using all samples: This is a question of gaining optimal power, not of ensuring correctness of inference. As the present paper does not claim to be the final word on the topic, such ways of optimizing inferential power are arguably beyond its scope, so we can leave this discussion to some future work.

Reviewer's Responses to Questions

**Comments to the Authors:**

Reviewer #1: The authors have done a good job of responding to the previous comments and suggestions. However, two responses need further clarifications.

1. Comparisons to propensity score matching (PSM). The sensitivity analysis suggests that PSM seems to generate results consistent with those from matching on covariates while tends to get more matching pairs and thus lead to smaller approximate p-values. This comparison indicates PSM might be a better approach for matching. However, in the revised Discussion section, the authors claim that the proposed approach is favored over PSM since "unconfoundedness" should be prioritized. I'm not completely clear why matching on covariates directly would achieve better unconfoundedness compared to PSM, especially considering the fact that a propensity score model can incorporate high dimensional potential confounders which to me appears as a more flexible tool to adjust for confounding. The authors should clarify more on this point.

2. I still believe some form of simulations should be done in order to validate the proposed approach. As the authors pointed out, understanding the causal effects of microbiome is nearly untapped and we cannot easily transfer our knowledge of the statistical properties of matching algorithms in regular univariate outcome scenarios to microbiome data. I understand that the goal of the approach is to provide exploratory analysis and hard thresholding of p-values for decision making is somewhat questionable. But we at least need to know whether the p-values from the approach under the null has expected behaviors (e.g., uniform distribution) and whether the approach after multiple correction has sufficient power to detect causal effects when the data generation process is known (e.g. in a simulation study). Even a simple low-dimensional example with several homogeneous treatment effect would make the paper much stronger.

Reviewer #2: See attached.

Reviewer #3: The authors made a great effort to address reviewers’ comments, but the authors' responses are not quite satisfactory.

1. In causal inference for observational studies, "sensitivity analysis" typically refers to a method that assesses the magnitude of violations from unconfoundedness (See Imbens & Rubin, 2015). The propensity score matching is another way of matching and needs an assessment of potentially unmeasured confounding effects, like every method in causal inference for observational studies. An assessment of unconfoundedness should be included, and it would make this paper more appealing.

2. The authors responded that the covariate adjustments method is unreliable, citing several references. However, the covariate adjustments method is another common approach used in causal inference (Pearl, 2000); the combination of this method with an inverse propensity scores weighting (known as the doubly robust estimator) is one of the most popular methods used in causal inference. Could the authors demonstrate its unreliability empirically? The estimates of the covariate adjustment method can also be given a causal interpretation under the same assumption of no unmeasured confounding effect. So, it is important to demonstrate matching, which discards a large portion of samples, is more reliable than the covariate adjustments method, which uses all samples.

3. Response to the authors' question about column titles of Table 1 (now Table 4):

In the Characteristics of Study Population section, a paragraph says that the number of matched pairs is 99 for the air pollution reduction experiment and 271 for the smoking prevention experiment. However, the sum of F and M is 271 in the left column (titled Air Pollution) and 99 in the right one (titled Smoking). I am not sure if only these numbers were switched or the column names were switched. I assumed the latter.

**Have the authors made all data and (if applicable) computational code underlying the findings in their manuscript fully available?**

Reviewer #1: Yes

Reviewer #2: Yes

Reviewer #3: Yes

PLOS authors have the option to publish the peer review history of their article (what does this mean?). If published, this will include your full peer review and any attached files.

Reviewer #1: No

Reviewer #2: No

Reviewer #3: No
---

## [Editor Report · Decision Letter 2]

21 Mar 2022

Dear Dr. Mueller,

We are pleased to inform you that your manuscript 'A randomization-based causal inference framework for uncovering environmental exposure effects on human gut microbiota' has been provisionally accepted for publication in PLOS Computational Biology.

Best regards,

Simon Anders

Guest Editor

PLOS Computational Biology

Kiran Patil

Deputy Editor

PLOS Computational Biology

Gues Editor's Comment:

The remaining reviewer concerns in the second review round did not concern technical aspects but rather the reviewers' impression that we authors overstated certain points. The authors have chosen one of several possible approaches in the field of causal inference and demonstrated how it can be used for metagenomic data. They could have chosen another. The reviewers were concerned that the authors wanted to claim that their choice is not just a possible choice but a superior one, but the authors made it clear in their response that it was not their intention to make any statement in this regard. i feel the the text in its current form no longer gives that impression, and the reviewers remaining concerns are addressed. Hence, I do not think that another review round is required and consider the paper as ready for publication. I hope the reviewers agree.

---

## [Editor Report · Acceptance letter]

24 Apr 2022

PCOMPBIOL-D-21-01632R2 

A randomization-based causal inference framework for uncovering environmental exposure effects on human gut microbiota

Dear Dr Müller,

I am pleased to inform you that your manuscript has been formally accepted for publication in PLOS Computational Biology. Your manuscript is now with our production department and you will be notified of the publication date in due course.

With kind regards,

Anita Estes
